# CONTROLLABLE EXPLORATION IN HYBRID-POLICY RLVR FOR MULTI-MODAL REASONING

**Zhuoxu Huang**[1][*] **Mengxi Jia**[2][*]**, Hao Sun**[2]**, Xuelong Li**[2][†]**, Jungong Han**[3,1][†]
[1]Aberystwyth University, [2]Institute of Artificial Intelligence (TeleAI), China Telecom, [3]Tsinghua University
zhh6@aber.ac.uk

## ABSTRACT

Reinforcement Learning with verifiable rewards (RLVR) has emerged as a primary learning paradigm for enhancing the reasoning capabilities of multi-modal large language models (MLLMs). However, during RL training, the enormous state space of MLLM and sparse rewards often leads to entropy collapse, policy degradation, or over-exploitation of suboptimal behaviors. This necessitates an exploration strategy that maintains productive stochasticity while avoiding the drawbacks of uncontrolled random sampling, yielding inefficient exploration. In this paper, we propose **CalibRL**, a hybrid-policy RLVR framework that supports controllable exploration with expert guidance, enabled by two key mechanisms. First, a distribution-aware advantage weighting scales updates by group rareness to calibrate the distribution, therefore preserving exploration. Meanwhile, the asymmetric activation function (LeakyReLU) leverages the expert knowledge as a calibration baseline to moderate overconfident updates while preserving their corrective direction. CalibRL increases policy entropy in a guided manner and clarifies the target distribution by estimating the on-policy distribution through online sampling. Updates are driven by these informative behaviors, avoiding convergence to erroneous patterns. Importantly, these designs help alleviate the distributional mismatch between the model's policy and expert trajectories, thereby achieving a more stable balance between exploration and exploitation. Extensive experiments across eight benchmarks, including both in-domain and out-of-domain settings, demonstrate consistent improvements, validating the effectiveness of our controllable hybrid-policy RLVR training. Code is available at https://github.com/zhh6425/CalibRL.

## 1 INTRODUCTION

Recent Large Language Models (LLMs), such as OpenAI-o1 (Jaech et al., 2024), DeepSeek-R1 (Guo et al., 2025), and Kimi-1.5 (Team et al., 2025), have achieved remarkable breakthroughs in complex reasoning by leveraging extended Chain-of-Thought (CoT) reasoning (Wei et al., 2022), showcasing unprecedented proficiency in multi-step logical inference. Building on these advances, Multi-Modal Large Language Models (MLLMs), including Virgo (Du et al., 2025), InternVL3 (Zhu et al., 2025), MiMo-VL (Xiaomi, 2025), and Ovis2.5 (Lu et al., 2025), have further extended reasoning into multi-modal domains, enabling complex visual reasoning, mathematical diagram interpretation, and cross-modal logical inference.

The success of recent models is largely attributed to advanced Reinforcement Learning using Verifiable Rewards (RLVR). Despite this progress, recent studies have shown fundamental challenges: improvements in policy performance often come at the cost of reduced policy entropy, creating a bottleneck where entropy depletion constrains further progress (Cui et al., 2025). Conventional RL methods usually incorporate entropy regularization (Liu et al., 2025a; Starnes et al., 2023) to encourage stochasticity, but the resulting high-entropy strategies rely on unguided random sampling, leading to inefficient exploration. This issue is especially pronounced in the enormous state space of MLLMs and significantly limits learning efficiency. Recently, combining Supervised Fine-Tuning

---

[*]Equal contribution.
[†]Corresponding author: Jungong Han & Xuelong Li

(SFT) with Reinforcement Learning (RL) training enables the integration of expert knowledge with self-improvement mechanisms, which can increase policy exploration while providing a clearer target distribution. Yet, neither the popular "SFT-then-RL" pipeline nor recent hybrid-policy frameworks that embed SFT supervision directly into RL training (Yan et al., 2025; Ma et al., 2025; Dong et al., 2025; Zhang et al., 2025) could provide stable and controllable exploration. In the sequential "SFT-then-RL" paradigm, the initial SFT stage anchors the policy to a static demonstration distribution, thereby diminishing its exploratory tendency in subsequent RL training. As a result, the policy struggles to adapt beyond the supervised baseline and fails to concentrate probability mass on novel, high-reward behaviors. Hybrid-policy methods that inject SFT supervision into RL suffer from a distributional mismatch between the current policy and expert trajectories. This mismatch introduces high bias and variance, leading to unstable policy learning. The resulting instability interferes with exploration and accelerates entropy collapse, driving the policy toward either overly deterministic or excessively random behaviors.

To address this dilemma, we propose **CalibRL**: *Hybrid-Policy RLVR with Controllable Exploration*, a framework that redefines the role of expert supervision as the calibration baseline. CalibRL treats expert data as a distributional baseline—a reference against which the model's on-policy behaviors are evaluated. From this perspective, our framework performs distributional calibration, explicitly designed to maintain sufficient policy entropy while guiding effective exploration. Responses are assessed relative to the baseline distribution from the expert: underrepresented yet correct reasoning paths are selectively reinforced, maintaining rare but informative behaviors as valuable exploration signals, while overconfident but erroneous predictions are penalized more strongly to prevent misleading convergence. This calibrated treatment transforms expert supervision from a rigid imitation signal into a nuanced guidance mechanism that balances exploration and performance, ensuring that policy entropy is retained while exploration is directed toward meaningful reasoning behaviors.

Crucially, controllable exploration is achieved by using two complementary mechanisms. An **advantage weighting** serves as an indicator of the relative likelihood of a response within its group, naturally emphasizing the contribution of the rare sample to enforce distribution calibration. In parallel, a **asymmetric activation** based on LeakyReLU leverages expert knowledge as a calibrated baseline to moderate overconfident updates while preserving their corrective direction, ensuring exploration remains guided and stable. Together, these mechanisms regulate both the strength and direction of policy updates, enabling guided and stable exploration, ensuring expert supervision remains informative rather than restrictive. We conduct extensive experiments across eight benchmarks and demonstrate consistent superiority over previous hybrid-policy methods, which fail to improve upon the GRPO baseline in our multi-modal scenario. In summary, the contributions of this work are threefold:

- We propose *CalibRL*, a hybrid-policy RLVR framework for controllable exploration in reasoning-oriented MLLMs, which leverages expert guidance to stabilize policy updates and increase policy entropy in a guided manner.

- Our CalibRL introduces two complementary mechanisms: *advantage weighting*, which emphasizes rare responses to enforce distribution calibration, and a *LeakyReLU–based asymmetric activation*, which moderates overconfident updates while preserving their corrective direction.

- We validate our method through extensive experiments on eight reasoning benchmarks, demonstrating substantial improvements over GRPO and state-of-the-art hybrid-policy baselines, with consistent gains across both in-domain and out-of-domain tasks.

## 2 PRELIMINARIES AND RELATED WORKS

This section introduces preliminary knowledge related to our work to facilitate understanding of the proposed method. We then review the most pertinent related work, identifying key limitations and outlining our research motivation.

### 2.1 REINFORCEMENT LEARNING WITH VERIFIABLE REWARDS (RLVR)

We start with definitions for training the LLM with RLVR. Given an input prompt $q$, and an LLM with parameters $\theta$, the reasoning generation task with the LLM is framed as a Markov Decision

Process (MDP) (Puterman, 2014). The LLM is represented as a policy $\pi_\theta$. The output response from the LLM is represented as the trajectory $\tau = (o_1, o_2, \ldots, o_T)$, where $T$ is the response length.

At each generation step $t \in [1, T]$, the MDP defines a state $s_t = (q, o_1, o_2, \ldots, o_{t-1})$, which represents the concatenation of the prompt $q$ and the response tokens generated up to step $t - 1$. The initial state $s_0 = q$. For each generation step $t$, the MDP defines an action $a_t = o_t \sim \pi_\theta(\cdot|s_t)$, representing the selection of $o_t$ as the next token with state $s_t$ as a condition.

In the circumstances, the policy $\pi_\theta$ provides a probability distribution over the LLM's vocabulary for the next token prediction. With a spare verifiable reward $R$ formulated as follows:

$$R(\tau) = \begin{cases} 1 & \text{if the response } \tau \text{ is correct,} \\ 0 & \text{otherwise.} \end{cases} \tag{1}$$

The training objective of RLVR is then formulated as follows:

$$\mathcal{J}(\theta) = \mathbb{E}_{q \sim \mathcal{D}, \tau \sim \pi_\theta(\cdot|q)}[R(\tau)], \tag{2}$$

where $\mathcal{D}$ is the prompts set. This objective is optimized with policy gradient optimization that aims to maximize the reward $R$ over the prompts $\mathcal{D}$.

In recent applications, the Group Relative Policy Optimization (GRPO) (Shao et al., 2024) has become the de facto choice owing to the success of Deepseek-R1 (Guo et al., 2025). The primary advantage stems from utilizing intra-group reward comparisons to derive the advantage for each trajectory. Given $G$ responses for each prompt $q$, the response group $\{\tau_i\}_{i=1}^G$ is validated by the reward function and used to calculate the advantages. This process can be formulated as follows:

$$\hat{A}_{i,t} = \frac{R(\tau_i) - mean(R(\{\tau_i\}_{i=1}^G))}{std(R(\{\tau_i\}_{i=1}^G))}. \tag{3}$$

The advantage signal guides the preference policy optimization, directing updates to increase the log-probability of tokens that exhibit high advantage values using a clipped PPO-style (Schulman et al., 2017) objective:

$$\mathcal{J}_{GRPO}(\theta) = \mathbb{E}_{q \sim \mathcal{D}, \tau \sim \pi_\theta(\cdot|q)} \left[ \sum_{t=1}^{|\tau|} \min \left( r_{i,t}(\theta) \hat{A}_{i,t}, \right.\right.$$

$$\left.\left. \text{clip}\left(r_{i,t}(\theta), 1 - \epsilon, 1 + \epsilon\right) \hat{A}_{i,t} \right) \right] - \beta \mathbf{D}_{KL}(\pi_\theta || \pi_{ref}), \tag{4}$$

where the $r_{i,t}(\theta) = \frac{\pi_\theta(\tau_{i,t}|s_{i,t})}{\pi_{\theta_{old}}(\tau_{i,t}|s_{i,t})}$ represent the importance sampling ratio. The KL-divergence penalty $\beta \mathbf{D}_{KL}(\pi_\theta || \pi_{ref})$ in the original GRPO (Shao et al., 2024) is used to regularize the policy update. In recent implementations (Yan et al., 2025; Dong et al., 2025), this KL term is usually omitted when training models for long CoT reasoning, as the model's distribution is expected to diverge significantly from the initial policy, rendering this constraint unnecessary.

## 2.2 RELATED WORK

**Reinforcement Learning for Reasoning Models.** Recent advances in reinforcement learning have significantly improved the reasoning ability of large language models (LLMs) (Guo et al., 2025; Jaech et al., 2024; Team et al., 2025; Xiaomi, 2025; Du et al., 2025; Zhu et al., 2025; Lu et al., 2025; Wang et al., 2025). These advancements have largely benefited from the emergence of Reinforcement Learning with Verifiable Rewards (RLVR) frameworks (Schulman et al., 2017; Shao et al., 2024; Liu et al., 2025b; Rafailov et al., 2023; Hao et al., 2025; Hu et al., 2025), which leverage verifiable signals to provide precise and stable reward shaping. Among these, the GRPO framework (Shao et al., 2024) has become particularly influential, introducing group-normalized advantage estimation as an efficient alternative to PPO (Schulman et al., 2017) under verifiable reward settings in mathematical reasoning. Despite these successes, recent studies have highlighted important limitations of on-policy RLVR when applied to reasoning tasks. Zhao et al. (2025) show that RL post-training often amplifies behaviors inherited from pre-training rather than fundamentally expanding reasoning capacity, leading to an "echo chamber" effect. Yue et al. (2025) further demonstrate that while RL improves sample efficiency (e.g., pass@1), it does not substantially broaden the model's reasoning boundary beyond that of the base model. Complementarily, Cui et al. (2025) provide an

analysis of entropy dynamics, revealing that on-policy updates tend to concentrate probability mass on a narrow set of high-reward trajectories, causing premature entropy collapse and limiting exploration. Together, these findings suggest that although RLVR stabilizes training through verifiable feedback, it also risks over-constraining the policy and failing to sustain the exploration necessary for discovering genuinely novel reasoning strategies.

**Hybrid-Policy Optimization Frameworks.** To address the limitations of purely on-policy RL, hybrid-policy optimization methods have been proposed, combining reinforcement learning with supervised fine-tuning (SFT) on expert data (Yan et al., 2025; Dong et al., 2025; Wu et al., 2025; Ma et al., 2025; Zhang et al., 2025). LUFFY (Yan et al., 2025) introduces off-policy guidance via mixed-policy optimization with regularized importance sampling, while RL-PLUS (Dong et al., 2025) develops a multi-importance sampling strategy combined with exploration-based advantage shaping. Other approaches, such as ReLIFT (Ma et al., 2025), interleave SFT and RL updates or introduce template-augmented objectives, whereas CHORD (Zhang et al., 2025) employs phased training to balance exploration and exploitation. Despite their innovations, these hybrid frameworks commonly rely on direct log-likelihood maximization of expert data, which imposes a unidirectional optimization pressure toward expert distributions. This often accelerates entropy collapse by suppressing alternative responses, thereby constraining policy diversity and weakening exploration. These limitations motivate our work: rather than treating expert data as an absolute imitation target, we propose to reinterpret it as a distributional baseline, enabling relative calibration of on-policy behaviors in a way that sustains entropy while still providing directional guidance.

# 3 METHOD

In this section, we reveal the entropy collapse acceleration process caused by the distribution gap between the expert data and the model behaviors. We then illustrate our controllable exploration under expert guidance, analyzing how it preserves policy entropy and directs exploration toward reliable behaviors.

## 3.1 LIMITATIONS OF DIRECT EXPERT OPTIMIZATION

Given a dataset of expert demonstrations $\mathcal{D} = \{(q_i, \tau_i^{\text{expert}})\}$, policies are typically trained to imitate expert behaviors by minimizing the negative log-likelihood, which is formalized as follows:

$$\mathcal{L}_{\text{expert}} = -\mathbb{E}_{(q_i, \tau_i^{\text{expert}}) \sim \mathcal{D}} \left[ \log \pi_\theta(\tau_i^{\text{expert}} | q_i) \right]. \tag{5}$$

The corresponding gradient update can be formalized as:

$$\nabla_\theta \mathcal{L}_{\text{expert}} = -\mathbb{E}_{(q_i, \tau_i^{\text{expert}})} \left[ \nabla_\theta \log \pi_\theta(\tau_i^{\text{expert}} | q_i) \right], \tag{6}$$

which monotonically increases $\pi_\theta(\tau^{\text{expert}} | q)$. This objective enforces unidirectional expert optimization: probability mass is consistently shifted toward expert responses, regardless of the model's current distribution. While this secures alignment, it also narrows distributional support and suppresses diversity of outputs.

In the sequential SFT-then-RL paradigm, this effect anchors the policy close to the expert distribution after SFT. Subsequent RL updates are further restricted by importance sampling and ratio clipping (Equation 4), which penalize deviations from the SFT initialization. Consequently, exploration remains confined to the neighborhood of expert behaviors, hindering adaptation to reward signals and limiting the discovery of higher-reward reasoning paths (Figure 1).

Hybrid-policy frameworks embed expert supervision directly into the RL objective, effectively rewarding higher likelihood of expert responses. While this guarantees continual expert alignment, normalization dictates that increasing $\pi_\theta(\tau^{\text{expert}} | q)$ necessarily decreases $\pi_\theta(\tau \neq \tau^{\text{expert}} | q)$, leading to overall entropy reduction, thereby accelerating entropy collapse and pushing the model toward overly deterministic expert-like behaviors:

$$\mathcal{H}(\pi_\theta(\cdot | q)) = -\sum_\tau \pi_\theta(\tau | q) \log \pi_\theta(\tau | q) \quad \downarrow. \tag{7}$$

In summary, both SFT-then-RL and hybrid-policy frameworks inherit the limitation of unidirectional expert optimization. By uniformly shifting probability mass toward expert trajectories, they

constrain exploration, accelerate entropy decay, and fail to adaptively account for the model's heterogeneous cognitive states. To address this limitation, we advocate a **relative calibration** perspective, treating expert data not as absolute imitation targets but as distributional baselines for evaluating on-policy behaviors.

## 3.2 Controllable Exploration under Expert Guidance

Our framework redefines the role of expert supervision by treating it as a distributional baseline rather than a strict imitation target. Given an input $q_i$, a model-generated response $\tau_i^{\text{policy}}$, and the corresponding expert response $\tau_i^{\text{expert}}$, we first define a log-probability gap as

$$\Delta \ell_i = \log \pi_\theta(\tau_i^{\text{policy}}|q_i) - \log \pi_\theta(\tau_i^{\text{expert}}|q_i) = \log \frac{\pi_\theta(\tau_i^{\text{policy}}|q_i)}{\pi_\theta(\tau_i^{\text{expert}}|q_i)}. \tag{8}$$

The quantity $\Delta \ell_i$ captures the model's relative preference between its own response and the expert's. A positive value indicates the model favors its own answer over the expert's, while a negative value indicates underconfidence relative to the expert.

We introduce a correctness signal $s_i = +1$ for correct responses and $s_i = -1$ for incorrect responses, as the actual reward, including the format reward, can exceed the $[0, 1]$ range. To ensure a rigorous definition that remains valid across a broad range of scenarios, we explicitly define the separate correctness signal $s_i$ rather than relying solely on a normalized version of the actual reward. We then design an objective that enables controllable exploration through an asymmetric activation based on the LeakyReLU operator, and an advantage weighting that emphasizes rare but informative events. The objective is written as

$$\mathcal{L}_{\text{exploration}} = |\hat{A}_i| \cdot \text{LeakyReLU}\big(-s_i \cdot \Delta \ell_i, \alpha\big), \tag{9}$$

where $|\hat{A}_i|$ is the absolute value of the group-wise advantage $\hat{A}_i$, representing an advantage weight capturing group-wise rarity. Multiplying $s_i$ with $\Delta \ell_i$ ensures that the optimization direction always aligns with correctness.

Under this objective, correct responses that are underweighted relative to the expert are reinforced, thereby broadening the support of the policy distribution and increasing entropy, while incorrect responses that are overweighted are suppressed, thereby shifting probability mass away from them and preventing entropy collapse. The LeakyReLU introduces asymmetric gradient gating, which allows the model to amplify underrepresented reasoning patterns while reducing overconfident predictions. Once a response's probability crosses the expert baseline, the slope parameter $\alpha \in (0, 1)$ controls the degree of further reinforcement or suppression, enabling exploration that is both directed and regulated. In this way, expert supervision functions as a relative reference rather than an absolute target, guiding probability redistribution in a calibrated manner while preserving sufficient stochasticity for the model to explore novel reasoning strategies.

The weighting $|\hat{A}_i|$ also contributes to controllable exploration by calibrating the scale of the update according to group-wise rarity. Large values occur when a rare correct response emerges among mostly incorrect ones, amplifying its reinforcement as an exploration signal. Similarly, when a rare incorrect response appears among mostly correct ones, the weighting increases its suppression. By modulating the update magnitude in this way, the model emphasizes rare but informative deviations while damping misleading outliers, ensuring that exploration remains selective and controllable.

We then integrate the controllable exploration term into the GRPO objective to obtain the final training objective

$$\mathcal{J}(\theta) = \mathbb{E}_{q \sim \mathcal{D}, \tau \sim \pi_\theta(\cdot|q)} \left[ \sum_{t=1}^{|\tau|} \min\Big( r_{i,t}(\theta)\hat{A}_{i,t}, \right.$$

$$\left. \text{clip}\left((r_{i,t}(\theta), 1 - \epsilon, 1 + \epsilon\right)\hat{A}_{i,t}\right) - \lambda \mathcal{L}_{\text{exploration}} \right], \tag{10}$$

where $\lambda$ balances standard PPO-style policy optimization and expert-guided exploration.

In summary, our controllable exploration framework combines correctness signals, asymmetric gradient gating via the LeakyReLU, and advantage weighting to modulate probability updates, selectively reinforcing underrepresented correct responses and suppressing overconfident errors while maintaining calibrated stochasticity.

## 4 EXPERIMENTS

### 4.1 SETUP

**Training dataset.** We construct our training dataset from the ViRL39K collection (Wang et al., 2025). Specifically, we sample geometry problems from ViRL39K and generate detailed Chain-of-Thought (CoT) responses using GPT-4o (Hurst et al., 2024). We then validate these responses across three criteria: correctness, format adherence, and logical coherence. This process yields 9,695 high-quality question-response pairs for training and 933 samples for validation. Notably, the validation set comprises samples that failed our CoT validation criteria, representing challenging cases where even GPT-4o struggled to produce satisfactory responses. We refer readers to the appendix for further training details.

**Benchmarks and Baselines.** We evaluate our method on a diverse suite of benchmarks covering both in-domain and out-of-domain (OOD) settings. For in-domain evaluation, we use the Geo3K (Lu et al., 2021) and GeoQA (Chen et al., 2021) test sets, along with our self-constructed GeoEval benchmark that failed the CoT validation criteria. To examine generalization, we test on MathVerse (Zhang et al., 2024), MathVision (Wang et al., 2024), and MathVista (Lu et al., 2024), which involve broader mathematical reasoning and visual understanding tasks. We further incorporate samples from ViRL39K (Wang et al., 2025), which covers diverse fields such as Science (Physics, Chemistry, Biology) and Spatial Reasoning. We also involve the MMMU (Yue et al., 2024) and the ScienceQA Lu et al. (2022). These benchmarks enable us to evaluate not only the effectiveness of our approach in geometry/mathematics but also its robustness across broader general domains.

For baselines, we compare against several representative approaches. GRPO (under customized training settings following Yan et al. (2025) and Dong et al. (2025)) serves as the standard reinforcement learning baseline, while SFT+GRPO reflects the sequential paradigm of supervised fine-tuning followed by reinforcement learning, highlighting the issues of distribution shift and overly supervised constraints. We also include LUFFY (Yan et al., 2025), a state-of-the-art hybrid-policy optimization method, to test whether our approach better preserves entropy compared to direct expert integration, and RLPLUS (Dong et al., 2025), which adopts conservative update strategies to stabilize training and thus serves as a strong baseline. We reproduce those methods on our dataset, keeping all training settings consistent throughout the training process. Our experiments are mainly conducted with Qwen2.5-VL-7B (Bai et al., 2025) as base models. During evaluation, we use MATH-VERIFY (Kydlíček) to score the Geo3K, GeoQA, GeoEval, Science, and Spatial Reasoning benchmarks, while the benchmarks MathVerse, MathVision, and MathVista are evaluated with the open-source VLMEVALKIT (Duan et al., 2024).

### 4.2 MAIN RESULTS

We first examine the training dynamics in Figure 1, taking GRPO as the reference baseline. As seen in the figure, SFT-then-GRPO suffers from overly supervised constraints: although it starts with relatively high reward values, its entropy remains excessively high throughout training, reflecting a lack of meaningful exploration. As a result, the policy distribution quickly solidifies, reward learning stagnates, and performance remains consistently low. In contrast, RL-PLUS (Dong et al., 2025) exhibits an opposite pathology. The entropy collapses too rapidly in the early phase and converges early to a certain pattern. This suppresses exploration and makes later reward learning ineffective, which leads to suboptimal reward curves and declining accuracy. The training dynamic shows that aggressive entropy reduction hinders sustained improvement. As also seen in Table 1,

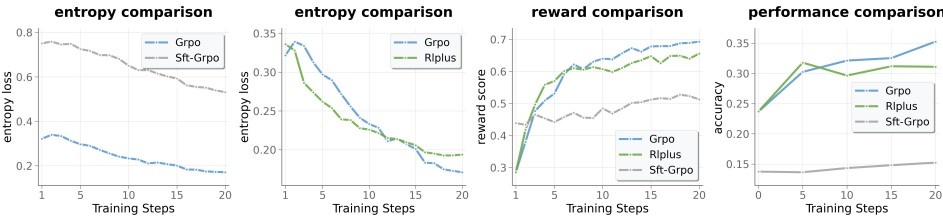

Figure 1: Entropy, reward, and accuracy curves of different methods. We split the entropy comparison into two panels for clarity.

both the hybrid-policy frameworks (Dong et al., 2025; Yan et al., 2025) fail to provide consistent benefits upon the GRPO baseline.

In contrast, quantitative results in Table 1 and Table 2 demonstrate our method's superiority over previous hybrid-policy methods. On in-domain geometry reasoning tasks, our CalibRL achieves an average performance gain of **5.45** percentage points over the GRPO baseline, substantially outperforming existing hybrid-policy methods, including LUFFY (↓0.84) and RL-PLUS (↓4.8). For out-of-domain reasoning benchmarks, we observe a consistent improvement of **2.61** percentage points over GRPO, while maintaining competitive advantages over LUFFY and RL-PLUS. Additionally, DAPO does not outperform GRPO in our setting. One possible explanation is that the higher clipping threshold leads to uncontrolled exploration, which further underscores the importance of our work. We further visualize the relative changes from the GRPO baseline in Figure 2. Our method consistently improves upon the GRPO framework across both in-domain and out-of-domain scenarios, whereas existing hybrid-policy approaches exhibit varying degrees of performance degradation, underscoring the effectiveness of our approach in overcoming their limitations.

Table 1: Performance comparison on in-domain geometry benchmarks.

| Method | in-distribution | out-of-distribution | | Avg. |
|---|---|---|---|---|
| | GeoEval | Geo3K | GeoQA | |
| GRPO | 26.15 | 39.77 | 52.52 | 39.48 |
| SFT+GRPO | 6.00 | 18.64 | 40.98 | 21.87 |
| DAPO | 25.19 | 40.93 | 52.52 | 39.55 |
| LUFFY (Yan et al., 2025) | 25.62 | 39.10 | 51.19 | 38.64 |
| RL-PLUS (Dong et al., 2025) | 20.69 | 36.94 | 46.42 | 34.68 |
| CalibRL | **33.44** | **40.60** | **60.74** | **44.93** |

Table 2: Performance comparison on out-of-domain benchmarks. We present the Science benchmark as 'Sci.' and the Spatial Reasoning benchmark as 'Sp.'.

| Method | Sci | Sp. | MathVerse | MathVision | MathVista | MMMU | ScienceQA | Avg. |
|---|---|---|---|---|---|---|---|---|
| GRPO | 61.99 | 48.02 | 49.01 | 27.89 | 70.00 | 55.44 | 88.34 | 57.24 |
| SFT+GRPO | 54.33 | 38.19 | 35.10 | 23.36 | 54.70 | 54.67 | 86.36 | 49.53 |
| DAPO | 61.61 | 50.21 | 47.82 | 27.07 | 70.40 | 54.67 | 87.95 | 57.10 |
| LUFFY (Yan et al., 2025) | 63.11 | 48.18 | 47.97 | 26.64 | 70.30 | 55.22 | 87.95 | 57.05 |
| RL-PLUS (Dong et al., 2025) | 61.21 | 47.55 | 46.40 | 27.14 | 69.90 | 55.88 | 87.50 | 56.51 |
| CalibRL | **65.12** | **53.64** | **51.35** | **27.93** | **71.90** | 56.55 | 89.04 | **59.36** |

Notably, the GeoEval validation set comprises samples that failed our CoT validation criteria, representing challenging cases where even GPT-4o struggled to produce satisfactory responses. The performance disparities on this demanding benchmark are particularly revealing: while the SFT+GRPO catastrophically fails with only 6.00% accuracy, and hybrid-policy methods struggle to match the GRPO baseline, our method achieves a remarkable 33.44% accuracy. This substantial improvement on such challenging instances demonstrates our method's superior capability in handling complex reasoning scenarios that typically confound existing approaches, validating the effectiveness of our proposed training strategy in maintaining robust performance on difficult edge cases.

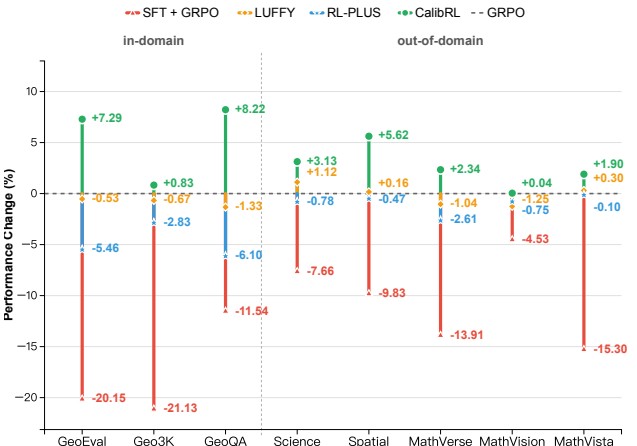

Figure 2: Performance comparison showing relative changes from GRPO baseline across in-domain geometry and out-of-domain reasoning tasks.

To validate the generalizability of our method across different model scales and architectures, we conduct experiments on both a smaller model (Qwen2.5VL-3B (Bai et al., 2025)) and a different

Table 3: Performance comparison on different base models.

| Method | GeoEval | Geo3K | GeoQA | MathVision | MathVista | Avg. |
|---|---|---|---|---|---|---|
| **Qwen2.5VL-3B** (Bai et al., 2025) | | | | | | |
| GRPO | 15.11 | 28.95 | 40.05 | 22.99 | 65.1 | 34.44 |
| LUFFY (Yan et al., 2025) | 12.54 | 28.95 | 33.55 | 23.42 | 62.9 | $32.27_{-2.17}$ |
| RL-PLUS (Dong et al., 2025) | 12.00 | 27.79 | 35.15 | 24.98 | 63.4 | $32.66_{-1.78}$ |
| CalibRL | 19.08 | 30.28 | 46.15 | 24.16 | 65.8 | $37.09_{+2.65}$ |
| **IntrenVL3-8B** (Zhu et al., 2025) | | | | | | |
| GRPO | 29.47 | 45.09 | 57.43 | 29.07 | 65.8 | 45.37 |
| LUFFY (Yan et al., 2025) | 13.83 | 15.14 | 48.67 | 29.07 | 62.0 | $33.74_{-11.63}$ |
| RL-PLUS (Dong et al., 2025) | 19.61 | 39.27 | 41.38 | 29.44 | 66.8 | $39.30_{-6.07}$ |
| CalibRL | 31.08 | 48.59 | 58.89 | 30.06 | 68.5 | $47.42_{+2.05}$ |

architecture (InternVL3-8B (Zhu et al., 2025)). As shown in Table 3, CalibRL achieves consistent improvements in both settings. On the smaller Qwen2.5VL-3B model, our method outperforms the GRPO baseline by 2.65 points, while other methods (LUFFY and RL-PLUS) show performance degradation of about 2 points. On the InternVL3-8B, CalibRL maintains its advantage with a 2.05 point improvement over GRPO, whereas competing methods suffer substantial drops. These results demonstrate that our controllable exploration mechanism generalizes effectively across varying models, consistently delivering performance gains while other recent approaches struggle with cross-model transferability.

## 4.3 ABLATION STUDIES

**Controllable Exploration.** We first conduct an ablation study to validate the importance of advantage weighting $|\hat{A}_i|$ by removing it from Equation 9, which treats all responses equally regardless of their distributional context. As shown in Table 4, this leads to substantial performance degradation across all benchmarks, demonstrating that advantage weighting is essential for our entropy control mechanism. The advantage weight $|\hat{A}_i|$ enables targeted distributional calibration by amplifying learning signals for rare correct responses while suppressing rare incorrect responses, systematically shifting probability mass toward underrepresented but valuable reasoning patterns. This represents the core of our entropy preservation strategy, ensuring controlled exploration that maintains distributional diversity. Without this mechanism, the exploration signal becomes indiscriminate, compromising both learning efficiency and generalization performance.

Table 4: Ablation studies on the controllable exploration objective. We present the Science benchmark as 'Sci.' and the Spatial Reasoning benchmark as 'Sp.'. The highlighted row represents our optimal results. **Bold** and underlined values denote the best and second-best results, respectively.

| Setting | in-domain | | | out-of-domain | | | | | Avg. |
|---|---|---|---|---|---|---|---|---|---|
| | GeoEval | Geo3K | GeoQA | Sci. | Sp. | MathVerse | MathVision | MathVista | |
| w/o $|\hat{A}_i|$ | 25.19 | 28.95 | 57.56 | 61.76 | 47.71 | 43.63 | 26.71 | 70.90 | 45.30 |
| w/ $|\hat{A}_i|$ | | | | | | | | | |
| $\alpha$=0.3 | 31.30 | 42.43 | 59.81 | 64.52 | 50.88 | 50.46 | **28.88** | 71.00 | 49.91 |
| $\alpha$=0.5 | **33.44** | 40.60 | **60.74** | 65.12 | **53.64** | **51.35** | 27.93 | **71.90** | **50.59** |
| $\alpha$=0.8 | 30.44 | **43.59** | 58.75 | **65.27** | 50.62 | 50.46 | 28.71 | 71.00 | 49.85 |

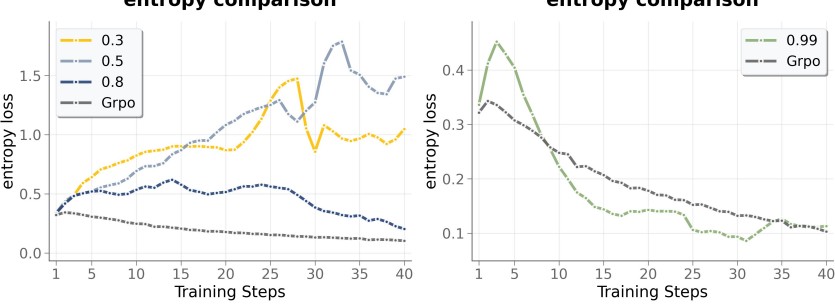

Figure 3: Entropy evolution during training for different $\alpha$ values in our framework. We split the comparison into two panels for clarity. The curves demonstrate how $\alpha$ controls exploration strength.

We further conduct an ablation study on the hyperparameter $\alpha$ in Equation 9 to evaluate its role in regulating exploration during training, with Table 4 reporting the performance across different $\alpha$ values and Figure 3 illustrating the corresponding entropy dynamics. Methodologically, $\alpha$ controls how the LeakyReLU mechanism scales the gradient of the exploration term. When the input to LeakyReLU is negative, the gradient is scaled by $\alpha < 1$, reducing the update strength and preventing excessive promotion or suppression. In this way, $\alpha$ regulates the balance between exploration and convergence in a controlled manner. The results confirm this principle. Small $\alpha$ (=0.3) values promote aggressive exploration in early training but result in unstable optimization characterized by entropy oscillations and premature decay, failing to maintain the sustained exploration necessary for effective learning. Large $\alpha$ values ($\geq$0.8) over-constrain the exploration signal, resulting in rapid entropy decay after a brief initial peak, thereby failing to maintain the distributional diversity essential for effective reasoning generalization. In contrast, an intermediate setting of $\alpha = 0.5$ achieves optimal exploration regulation by maintaining smooth entropy growth without destabilizing oscillations, reaching sustained high-level exploration that enables effective learning of diverse reasoning patterns while preserving training stability, ultimately resulting in the highest overall performance and wins on the majority of benchmarks.

These findings highlight that properly calibrated exploration control is essential for reasoning performance. More importantly, our framework enables fully controllable exploration regulation, allowing systematic discovery of optimal balance points and effective navigation toward superior reasoning solutions across varying task requirements.

**Balance Weight.** We then test the performance of our method with different balance weights $\lambda$, controlling the trade-off between standard policy optimization and expert-guided controllable exploration in Equation 11. We observe that small $\lambda$ values favor in-domain improvements but provide limited generalization, as the model tends to overfit to training-like trajectories. Conversely, very large $\lambda$ values suppress policy learning and hurt both in-domain and out-of-domain performance due to excessive reliance on expert supervision. A moderate balance achieves the best overall performance, yielding competitive in-domain results while substantially improving out-of-domain generalization.

Table 5: Ablation studies on the balance weight. We present the Science benchmark as 'Sci.' and the Spatial Reasoning benchmark as 'Sp.'. The highlighted row represents our optimal results. **Bold** and underlined values denote the best and second-best results, respectively.

| $\lambda$ | in-domain | | | out-of-domain | | | | | Avg. |
|---|---|---|---|---|---|---|---|---|---|
| | GeoEval | Geo3K | GeoQA | Sci. | Sp. | MathVerse | MathVision | MathVista | |
| 0.01 | 29.47 | 38.10 | 58.36 | 64.49 | 50.10 | 49.21 | **28.72** | 70.80 | 48.66 |
| 0.03 | 30.44 | **42.10** | **61.41** | 63.68 | 51.14 | 50.69 | 27.86 | **72.10** | 49.93 |
| 0.05 | 28.83 | 41.10 | 57.82 | 64.23 | 49.01 | 49.21 | 27.80 | 71.10 | 48.64 |
| 0.1 | **33.44** | 40.60 | 60.74 | **65.12** | **53.64** | **51.35** | 27.93 | 71.90 | **50.59** |
| 0.3 | 23.58 | 26.96 | 58.36 | 61.70 | 48.02 | 42.06 | 25.82 | 70.10 | 44.58 |
| 0.5 | 25.62 | 31.28 | 55.57 | 61.04 | 47.14 | 43.07 | 27.01 | 70.30 | 45.13 |

**Different entropy control.** We also conduct ablations to compare our controllable exploration with several entropy regularization methods. Results are reported in Table 6. In summary, **only CalibRL delivers consistent improvements** across all evaluated benchmarks. We first compare our method with fixed-coefficient entropy regularization by setting the entropy coefficient to 0.01 according to (Yan et al., 2025; Dong et al., 2025), which results in a degradation in performance. We then apply the widely used entropy-control mechanisms based on KL and clip-covariance regularization (Cui et al., 2025) on top of GRPO. The KL-Cov variant provides a slight improvement on some tasks but remains noticeably weaker than our CalibRL, while the Clip-Cov variant again results in a performance drop. Compared with conventional entropy-based methods, CalibRL enhances policy entropy in a guided manner that avoids unguided randomness and directs exploration toward meaningful reasoning behavior. As a result, it achieves more effective exploration than the compared entropy-based baselines.

**Different activation functions.** Adopting the leaky mechanism of the LeakyReLU function enables us to build a controllable entropy–shaping mechanism, facilitated by its adjustable negative slope. We show the necessity of this design through ablations using other activation functions, where the ReLU, sigmoid, and Huber fully truncate negative values, provide no controllable scaling, and the

Table 6: Performance comparison of different entropy control.

| Method | in-distribution | out-of-distribution | | Avg. |
|---|---|---|---|---|
| | GeoEval | Geo3K | GeoQA | |
| GRPO | 26.15 | 39.77 | 52.52 | 39.48 |
| + entropy coefficient = 0.01 | 22.62 | 40.27 | 47.48 | 36.79 |
| + KL-Cov (Cui et al., 2025) | 26.47 | 41.60 | 53.58 | 40.55 |
| + Clip-Cov (Cui et al., 2025) | 25.40 | 39.43 | 52.12 | 38.98 |
| CalibRL | 33.44 | 40.60 | 60.74 | 44.93 |

tanh, which cannot modulate the magnitude, none of which offer the required level of control. Results are reported in Table 7, the three ReLU, sigmoid, and Huber either failed to improve the GRPO baseline or only provided a slight gain. On the other hand, the tanh provides a relatively strong improvement, showing the importance of the negative value in the entropy-shaping mechanism. Finally, our design for the CalibRL consistently achieved the highest performance.

Table 7: Ablations on activation functions.

| Method | in-distribution | out-of-distribution | | Avg. |
|---|---|---|---|---|
| | GeoEval | Geo3K | GeoQA | |
| GRPO | 26.15 | 39.77 | 52.52 | 39.48 |
| CalibRL | | | | |
| w/ ReLU | 26.47 | 33.11 | 57.96 | 39.18 |
| w/ Sigmoid | 26.69 | 32.61 | 60.08 | 39.79 |
| w/ Huber | 24.22 | 27.62 | 55.70 | 35.85 |
| w/ Tanh | 30.65 | 40.93 | 58.62 | 43.40 |
| w/ LeakyReLU($\alpha = 0.5$) | 33.44 | 40.60 | 60.74 | 44.93 |

**Different reference policies.** We conduct ablations to compare different reference policies for computing $\Delta \ell_i$. Specifically, we replace the $\log \pi_\theta(\tau_i^{\text{expert}}|q_i)$ with the reference policy $\log \pi_\theta(\tau_i^{\text{ref}}|q_i)$, resulting in a log-probability gap as: $\Delta \ell_i' = \log \pi_\theta(\tau_i^{\text{policy}}|q_i) - \log \pi_\theta(\tau_i^{\text{ref}}|q_i)$

Results are reported in Table 8. The expert baseline strongly suppresses the reference policy baseline, showing the importance of the expert guidance in our controllable exploration design.

Table 8: Ablations on reference baselines.

| Method | in-distribution | out-of-distribution | | Avg. |
|---|---|---|---|---|
| | GeoEval | Geo3K | GeoQA | |
| CalibRL w/ expert baseline | 33.44 | 40.60 | 60.74 | 44.93 |
| CalibRL w/ ref policy baseline | 27.87 | 40.77 | 55.57 | 42.74 |

## 5 CONCLUSION

In this work, we investigated the fundamental tension between exploration and supervision in training reasoning-oriented MLLMs and demonstrated that existing SFT-then-RL and hybrid-policy frameworks either remain overly constrained by supervised baselines or suffer from entropy collapse. To address this challenge, we proposed CalibRL: Hybrid-Policy RLVR with Controllable Exploration, a principled framework that reinterprets expert supervision as relative calibration rather than direct imitation, thereby preserving policy entropy while providing directional guidance. Through a LeakyReLU-based asymmetric activation and an advantage weighting mechanism, our method achieves explicit control over exploration, enabling the model to reinforce underrepresented yet correct reasoning trajectories while discouraging overconfident errors. Extensive experiments across eight benchmarks confirmed the effectiveness of our approach, showing consistent improvements over GRPO and strong hybrid-policy baselines. We believe this work provides a step toward more generalizable reasoning in MLLMs, highlighting the importance of controllable exploration as a key ingredient in future post-training strategies.

## 6 REPRODUCIBILITY STATEMENT

We have made extensive efforts to ensure the reproducibility of our work. Details of the experimental setup are provided in Section 4.1, and the full training configurations are documented in the Appendix (Training Settings). Additional implementation details are included in the supplementary materials. Our codebase is built upon the `verl` framework (Sheng et al., 2024), with the main implementations located in the `src/` directory. To further support reproducibility, we will release all training data and pretrained model weights upon publication.

### ACKNOWLEDGMENTS

This work was supported in part by National Natural Science Foundation of China No. 62441235, and is also supported by Beijing Natural Science Foundation (L257005).

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

## A  TRAINING PROMPT

As shown in Table 9, for a multi-modal image-question input, we concatenate the question with our instruction, which guides the model to perform step-by-step reasoning and specifies the desired output format.

Table 9: **Prompt template.**

<image> question.
You FIRST think about the reasoning process as an internal monologue and then provide the final answer.
The reasoning process MUST BE enclosed within <think> < /think> tags. The final answer MUST BE put in <answer> < /answer> tags.

## B  TRAINING SETTINGS.

We conduct all our experiments on 8 NVIDIA A800 80G GPUS. For fair comparison, we follow the standard GRPO training setup with several modifications as previous works (Liu et al., 2025b; Yan et al., 2025; Dong et al., 2025). First, we disable the KL regularization term by setting its coefficient to zero and removing both the length normalization and the standard error normalization in the original GRPO loss. For rollout generation, we use a temperature of 1.0 and perform 10 rollouts per prompt. In the case of hybrid-policy RL, we additionally include one off-policy rollout. The rollout batch size is set to 480, while the update batch size is 120. As the reward signal, we employ MATH-VERIFY (Kydlíček), combined with a lightweight format reward of 0.1. For our controllable exploration, we set the $\lambda$ weight to 0.1 as the default. The GeoEval data remained **completely unseen during training**. We trained all models for a fixed number of steps without any early stopping, and checkpoints for comparison were selected at identical training steps across all experiments.

## C  ADDITIONAL EXPERIMENTAL RESULTS

**General applicability beyond visual reasoning.** In this work, we focus on achieving stable entropy control in RLVR for VLMs, where the challenge becomes especially severe in the vast state space of MLLMs and substantially limits learning efficiency. While a series of works like LUFFY (Yan et al., 2025) and RL-PLUS (Dong et al., 2025) have achieved promising results for LLMs, such results have not transferred to MLLMs, leaving a notable gap in the literature, one that our work aims to fill.

However, our method can certainly be expanded to broader applications beyond visual reasoning. We extend CalibRL to pure text math reasoning tasks to demonstrate the effectiveness beyond visual reasoning. Specifically, we train Qwen2.5-VL-7B on a 9k-sample subset of the MATH dataset (Hendrycks et al., 2021) and evaluate the model on the in-distribution benchmark MATH-500 (Hendrycks et al., 2021) and the out-of-distribution benchmark AMC (Li et al., 2024).

Table 10: Performance of the Qwen2.5-VL-7B model on math reasoning tasks.

| Method | MATH-500 (in-distribution) | AMC (out-of-distribution) |
|---|---|---|
| Qwen2.5-VL-7B | 64.8 | 31.97 |
| GRPO | 68.5 | 31.59 |
| CalibRL (ours) | 70.2 | 32.08 |

Results are reported in Table 10. Training on pure text data, CalibRL yields stronger gains than GRPO on MATH-500, demonstrating more effective in-distribution improvement. Moreover, unlike GRPO, which fails to enhance out-of-distribution performance, CalibRL continues to deliver benefits on AMC, achieving strong generalization results.

**Scalability to larger model sizes.** We also verify our method on the Qwen2.5-VL-32B model. We train the Qwen2.5-VL-32B with GRPO and our CalibRL, respectively, using the same training data we constructed for our main paper results. Results are reported in Table 11. Our method exhibits consistent improvements when scaled to a larger model.

Table 11: Performance of the Qwen2.5-VL-32B model.

| Method | in-distribution | out-of-distribution | | Avg. |
|---|---|---|---|---|
| | GeoEval | Geo3K | GeoQA | |
| Qwen2.5-VL-32B | 37.94 | 48.24 | 60.87 | 48.60 |
| GRPO | 44.48 | 51.75 | 63.53 | 53.25 |
| CalibRL(ours) | 48.77 | 52.58 | 67.90 | 56.42 |

**Analysis of potential length bias in $\Delta\ell_i$.** We acknowledge that using the $\Delta\ell_i$ may introduce a mild length preference. However, in our method, the expert trajectory is intended to guide not only correctness but also the answering paradigm—including stylistic preferences such as avoiding overly long or excessively short responses. In this sense, a small length preference is aligned with our design goal of encouraging the model to follow the expert's response style.

Moreover, this bias does not override correctness learning. The policy is still optimized primarily by the main GRPO loss, which determines reward-maximizing behavior. The exploration term is scaled by a small $\lambda$ and serves only to regulate the degree of deviation from the expert, rather than to dictate correctness. Thus, the optimization direction remains dominated by the policy objective, and we did not observe harmful interference with correctness.

To further analyze the influence of such length bias, we conduct an ablation by adding a length normalization to the $\Delta\ell_i$ computation. Results are reported in Table 12. The strong performance of the model without length normalization supports our original design.

Table 12: Performance on length norm.

| Method | in-distribution | out-of-distribution | | Avg. |
|---|---|---|---|---|
| | GeoEval | Geo3K | GeoQA | |
| CalibRL | | | | |
| w/o length norm | 33.44 | 40.60 | 60.74 | 44.93 |
| w/ length norm | 29.26 | 39.10 | 60.34 | 42.90 |

**Effect of weaker expert baselines.** In our main results, we use GPT-4o as the expert to generate CoT responses for training. To further verify the generalizability of CalibRL, we additionally compare GRPO and CalibRL when the training data are produced by Qwen2.5-VL-72B, which serves as a weaker expert compared to GPT-4o. We keep the size of the Qwen-generated dataset comparable to the GPT-generated one (approximately 9k samples). However, because the two expert models differ in capability, the filtered sets of correct responses are not identical, and we cannot guarantee that both methods are trained on the same sampled data. Nonetheless, this setting still provides a meaningful evaluation of the robustness and generality of CalibRL under varying expert quality.

The results in Table 13 show that CalibRL consistently delivers substantial improvements over GRPO in both settings. As expected, the magnitude of improvement depends on the quality of the expert baseline, with a stronger expert providing a high-quality, generalizable, and informative reference. This highlights the importance of expert guidance in controllable-entropy RLVR and further demonstrates that CalibRL can effectively adapt its learning behavior to the quality of the expert model, leading to significant gains in training performance.

## D  COMPUTATIONAL COST AND SAMPLE EFFICIENCY

Our method adds one off-policy expert response per prompt to the standard GRPO group of G policy-generated responses (where G = 10 in our implementation). This introduces a well-defined

Table 13: Trained on different expert data.

| Method | in-distribution | out-of-distribution | | Avg. |
|---|---|---|---|---|
| | GeoEval | Geo3K | GeoQA | |
| On Qwen data | | | | |
| GRPO | 23.26 | 41.10 | 49.87 | 38.08 |
| CalibRL | 26.90 | 40.43 | 56.90 | 41.41 $_{+3.33}$ |
| On GPT data | | | | |
| GRPO | 26.15 | 39.77 | 52.52 | 39.48 |
| CalibRL | 33.44 | 40.60 | 60.74 | 44.93$_{+5.45}$ |

incremental cost that we quantify below (see Table 14). The expert response requires one additional forward pass to compute the expert log probability. Importantly, different from previous hybrid-policy methods, including LUFFY and RL-PLUS, the expert response **does not participate in gradient computation**. It serves only as a reference baseline. This means while we have G+1 forward passes per prompt, we still only perform G backward passes, **identical to standard GRPO**. Also, different from LUFFY, we require no additional reward for the expert trajectory, since we only include on-policy samples in the advantage calculation. Additionally, expert data collection is a one-time offline cost, not incurred during training, so rollout sampling overhead is zero.

Table 14: Per-prompt computational breakdown for a typical group size of G.

| Component | GRPO | CalibRL | Additional Cost |
|---|---|---|---|
| Rollout sampling | G | G | 0 |
| Forward passes | G | G + 1 | +1 |
| Reward model queries | G | G | 0 |
| Backward passes | G | G | 0 |

# E  THEORETICAL GROUNDING OF $|\hat{A}_i|$

**Possible double-counting of advantage terms.** Our training objective:

$$\mathcal{J}(\theta) = \mathbb{E}_{q \sim \mathcal{D}, \tau \sim \pi_\theta(\cdot|q)} \left[ \sum_{t=1}^{|\tau|} \min \left( r_{i,t}(\theta) \hat{A}_{i,t}, \right. \right.$$

$$\left. \left. \text{clip}\left(r_{i,t}(\theta), 1 - \epsilon, 1 + \epsilon\right) \hat{A}_{i,t} \right) - \lambda |\hat{A}_i| \cdot \text{LeakyReLU}\left( - s_i \Delta \ell_i, \alpha \right) \right]. \tag{11}$$

In the main GRPO objective, $\hat{A}_{i,t}$ determines the update direction of the policy, while in the exploration term, $|\hat{A}_i|$ is treated purely as a static importance weight. The two occurrences of the advantage influence different aspects of the optimization: the GRPO term governs policy improvement, whereas the exploration term modulates the strength of trajectory-level exploration correction.

The exploration loss gradient is:

$$\frac{\partial \mathcal{L}_{\text{exploration}}}{\partial \theta} = |\hat{A}_i| \cdot \text{LeakyReLU}'(-s_i \Delta \ell_i) \cdot (-s_i) \cdot \nabla_\theta \log \pi_\theta(a_i), \tag{12}$$

while the GRPO gradient is:

$$\frac{\partial}{\partial \theta} \left( r_{i,t}(\theta) \hat{A}_{i,t} \right) = \hat{A}_{i,t} \cdot \nabla_\theta r_{i,t}(\theta)$$

$$= \hat{A}_{i,t} \cdot r_{i,t}(\theta) \cdot \nabla_\theta \log \pi_\theta(a_{i,t}). \tag{13}$$

Both terms produce updates through ($\nabla_\theta \log \pi_\theta$), but with different multiplicative coefficients (the GRPO term uses ($\hat{A}_{i,t} r_{i,t}$) while the exploration term uses ($|\hat{A}_i|(-s_i \text{LeakyReLU}(\cdot))$)) and is scaled

by $\lambda$). The key observation is that these coefficients are additive in the total gradient, not multiplicative. The final gradient is:

$$\nabla_\theta \mathcal{J} = \sum_t \hat{A}_{i,t} \cdot r_{i,t}(\theta) \cdot \nabla_\theta \log \pi_\theta(a_{i,t}) - \lambda \cdot |\hat{A}_i| \cdot (-s_i) \cdot \text{LeakyReLU}(\cdot) \cdot \nabla_\theta \log \pi_\theta(a_i). \quad (14)$$

The advantage appears in two separate, additive gradient contributions that can constructively or destructively interfere depending on their signs, with the hyperparameter $\lambda$ controlling the relative magnitude of exploration correction versus policy improvement, preventing uncontrolled amplification. In our experiments, $\lambda = 0.1$ ensures the exploration term provides a measured correction signal without dominating the GRPO updates.

**The "rarity" interpretation of $|\hat{A}_i|$.** We provide a formal characterization of the intuition of using $|\hat{A}_i|$ value to naturally capture group-wise "rarity".

First, consider several example groups of sequence-level rewards:

- Group 1: [0,1,1,1], where the "0" answer is relatively rare within the group. The absolute value of the group advantage is [0.75,0.25,0.25,0.25]. The "0" has the largest $|\hat{A}|$, indicating that a reward of "0" is the group-wise outlier.
- Group 2: [1,0,0,0], where the "1" answer is relatively rare within the group. This group resulted in the same absolute value of the group advantage: [0.75,0.25,0.25,0.25], also indicating that a reward of "1" is "rare" in this group.

The examples demonstrate that the frequency of correct/incorrect answers and $|\hat{A}_i|$ correspond symmetrically and automatically, without requiring additional heuristics. We further show a continuous change between $|\hat{A}_i|$ and reward frequency in a group of 10 samples in Figure 4. The curve shows a strictly monotonic mapping between rarity and magnitude.

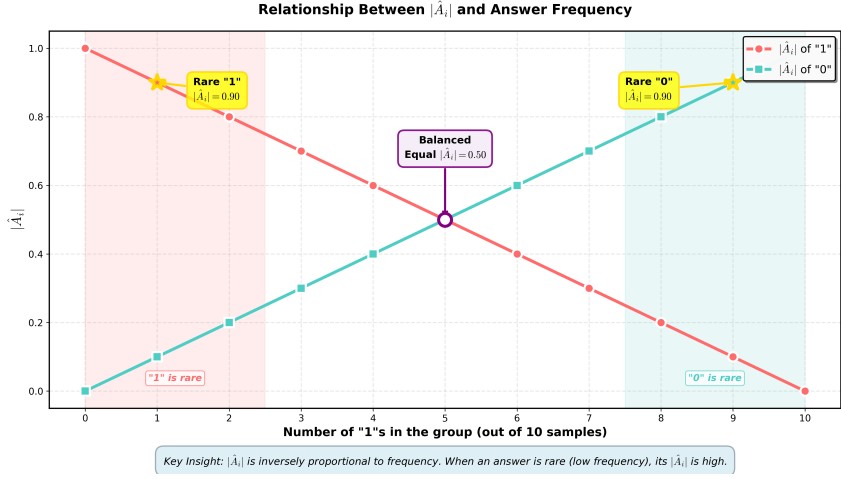

Figure 4: Relationship between $|\hat{A}_i|$ and reward frequency in a group of 10 samples.

Furthermore, to analyze robustness under small additive reward noise (e.g., formatting bonuses), consider the perturbed reward

$$R'(\tau_i) = R(\tau_i) + \delta_i, \qquad |\delta_i| \le \epsilon.$$

The corresponding advantage becomes

$$\hat{A}'_{i,t} = R'(\tau_i) - \text{mean}(R'(\{\tau_j\}_{j=1}^G)) = (R(\tau_i) - \mu_G) + (\delta_i - \bar{\delta}),$$

where

$$\mu_G = \text{mean}(R(\{\tau_j\}_{j=1}^G)), \qquad \bar{\delta} = \text{mean}(\{\delta_j\}_{j=1}^G).$$

Thus, the perturbation to the advantage is

$$\hat{A}'_{i,t} - \hat{A}_{i,t} = \delta_i - \bar{\delta},$$

which is a mean-centered noise term satisfying

$$|\delta_i - \bar{\delta}| \leq 2\epsilon.$$

Since typically

$$|R(\tau_i) - \mu_G| = O(1),$$

the ordering of $|\hat{A}_{i,t}|$, which determines group-wise rarity, is preserved for sufficiently small $\epsilon$. In the common case where noise is approximately uniform across samples (e.g., shared formatting bonuses), we have $\delta_i \approx \bar{\delta}$, and the perturbation cancels:

$$\hat{A}'_{i,t} \approx \hat{A}_{i,t}.$$

Hence, mean-centered normalization makes the rarity signal $|\hat{A}_{i,t}|$ inherently robust to small reward noise.

## F  EFFECTIVENESS ANALYSIS OF CALIBRL

We obtain several statistical data points from our trained checkpoints to contrast CalibRL with RL-PLUS and LUFFY and to reveal the structural differences brought by explicit calibration. Both LUFFY and RL-PLUS attempt to use experts without an effective calibration mechanism, which leads to either diluted expert signals or unstable confidence dynamics. In contrast, CalibRL introduces a calibration mechanism that stabilizes expert influence, as reflected in the following statistics.

As in Table 15, LUFFY collapses to a single expert mode. The policy and expert distributions become nearly indistinguishable after training. The model treats expert responses as a uniform style to imitate. It loses the ability to evaluate where expert guidance should matter and where exploration should continue. RL-PLUS shows the opposite tendency. The policy becomes prematurely certain. Expert responses also lose diversity and drift toward the policy. The model reinforces its own early preferences without checking them against a stable expert reference. It converges quickly but without reliable calibration. In both cases, expert information does not form a stable reference for exploration.

Differently, our CalibRL is not designed to directly learn or fit the expert distribution, which would resemble an off-policy imitation objective. Instead, CalibRL uses expert responses as a baseline that calibrates exploration rather than constrains generation. As shown in Equations 8 and 9, the distributional influence of expert samples is not a monotonic or uniform increase in consistency with the expert distribution. Specifically, when the policy's rollout produces an incorrect answer, optimization pushes the policy closer to the expert sample distribution; however, when the policy itself produces a correct rollout, the method suppresses the probability of expert samples. Thus, distributional shifts measured across all samples cannot directly reflect the calibration mechanism that CalibRL performs.

The statistics in Table 15 support this distinction. CalibRL maintains the broadest policy entropy, showing that the model stays exploratory rather than collapsing into a single expert mode as in LUFFY or becoming prematurely overconfident as in RL-PLUS. Expert responses also remain diverse and on-policy, which provides a stable reference for calibration rather than a target distribution to mimic. CalibRL shows a negative $\Delta\ell_i$, indicating that the model consistently assigns a higher likelihood to expert answers and uses this signal to navigate exploration toward better solutions. This calibrated behavior produces higher rewards and shorter, more precise outputs.

Table 15: Statistical data points from our trained checkpoints.

| Method | $\Delta\ell_i$ | policy entropy | reward | response length (expert length 382.19) |
|---|---|---|---|---|
| Qwen2.5-VL-7B | 0.1459 | 0.4401 | 0.2029 | 437.76 |
| GRPO | 0.4256 | 0.2508 | 0.5381 | 431.79 |
| LUFFY | 0.0881 | 0.4688 | 0.4980 | 414.58 |
| RL-PLUS | 0.3452 | 0.2580 | 0.4975 | 425.08 |
| CalibRL | -0.8025 | 1.4968 | 0.5667 | 271.71 |

## G  CASE STUDY

We select cases from the challenging MathVision (Wang et al., 2024) benchmark and compare our response with the GRPO baseline, the SFT+GRPO baseline, the LUFFY (Yan et al., 2025), and the RL-PLUS (Dong et al., 2025).

We first compare the case response among our CalibRL, the GRPO baseline, and the SFT+GRPO baseline in Figure 5. The GRPO baseline, lacking explicit guidance, exhibits erroneous reasoning and hallucinatory requirements (highlighted in red). In contrast, the SFT+GRPO baseline is overly constrained by the supervised signals, limiting its exploratory capacity; as a result, it attempts to frame the problem in a purely mathematical manner, which ultimately leads to ineffective reasoning. Distinct from both, our approach leverages effective guidance to explore the most efficient solution path, directly and precisely resolving the problem.

Similarly, in the case analysis shown in Figure 6, LUFFY falls into inefficient exploration, which results in both visual understanding errors and flawed reasoning (highlighted in red), ultimately failing to produce the correct answer. RL-PLUS leverages multiple importance sampling (Dong et al., 2025) to partially alleviate inefficient reasoning. However, it fails to fundamentally resolve the problem. In contrast, our method maintains effective exploration while preserving expert guidance, thereby successfully and reliably addressing the task.

## H  THE USE OF LARGE LANGUAGE MODELS

We employed large language models in our work to check the grammatical correctness of the paper and to provide refinements to selected sentences for improved clarity and readability.

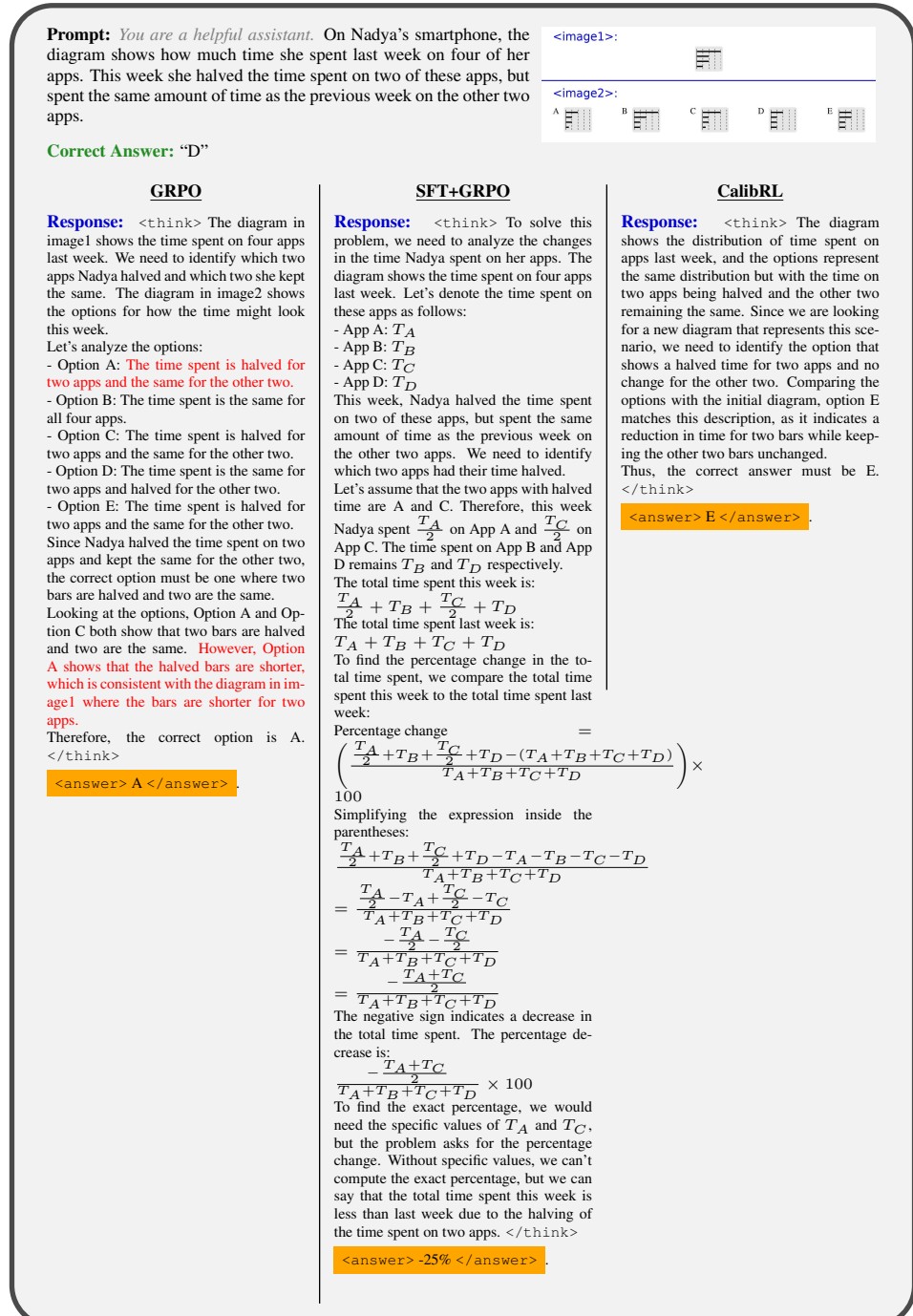

Figure 5: **A case of CalibRL compared with baselines GRPO and SFT+GRPO.**

**Prompt:** *You are a helpful assistant.* The diagram shows the floor plan of Renate's house. Renate enters her house from the terrace (Terrasse) and walks through every door of the house exactly once. Which room does she end up in?

**Correct Answer:** "2"

### LUFFY

**Response:** `<think>` To solve this problem, we need to determine which room Renate ends up in after walking through every door exactly once. This is essentially a problem of finding an Eulerian path in a graph, where each room is a node and each door is an edge.
Here's the step-by-step reasoning:
1. Identify the nodes (rooms) and edges (doors) in the graph.
2. Count the degree (number of edges) of each node.
3. According to Euler's theorem, a graph has an Eulerian path if and only if exactly zero or two nodes have an odd degree. All other nodes must have an even degree.
4. If there are exactly two nodes with odd degrees, the Eulerian path will start at one of those nodes and end at the other.
5. If there are no nodes with odd degrees, the Eulerian path can start at any node and end at any node.
In the given diagram:
- The rooms are labeled as 1, 2, 3, 4, and 5.
- The terrace is not a room but an entrance.
Let's count the degrees of each node:
- Room 1: 2 edges
- Room 2: 2 edges
- Room 3: 2 edges
- Room 4: 2 edges
- Room 5: 2 edges
All rooms have an even degree, which means the graph has an Eulerian path that can start at any room and end at any room.
Since Renate enters from the terrace, we can assume she starts at room 5 (since it's the only room connected to the terrace). If she walks through every door exactly once, she will end up in the room that is connected to the last door she used. In this case, she will end up in room 1, as she starts from room 5 and walks through each room exactly once, ending at room 1.
Therefore, the room she ends up in is room 1.`</think>`

`<answer>` 1 `</answer>` .

### RL-PLUS

**Response:** `<think>` To solve this problem, we need to determine which room Renate ends up in after walking through every door exactly once. This is a classic problem of finding an Eulerian path in a graph, where each room is a node and each door is an edge.
Let's analyze the floor plan:
- The house has five rooms labeled 1, 2, 3, 4, and 5.
- There are doors connecting these rooms.
- The terrace is not connected to the house, but Renate enters from it.
To find the room Renate ends up in, we need to identify the room that has an odd number of edges (doors) connected to it.
In an Eulerian path, the path starts at a vertex with an odd degree and ends at a vertex with an odd degree. All other vertices must have an even degree.
Let's count the edges for each room:
- Room 1: 3 edges
- Room 2: 3 edges
- Room 3: 3 edges
- Room 4: 3 edges
- Room 5: 2 edges
Room 5 has an even number of edges, while all other rooms have an odd number of edges. Therefore, Room 5 is the only room that can be the endpoint of the Eulerian path.
Thus, Renate ends up in Room 5. `</think>`

`<answer>` 5 `</answer>` .

### CalibRL

**Response:** `<think>` The floor plan represents a grid-like structure. To walk through every door exactly once, Renate must start at the terrace, walk through each room, and end at a room that allows her to do so without retracing her steps. The rooms must be traversed in such a way that they form a path that crosses each door once and only once. Since this is a 2x3 grid, one room will not have an exit corresponding to a non-visited door. Here, room 2 appears to be the room without an exit corresponding to an unvisited door, and it is connected to three doors. The logical end point is room 2. `</think>`

`<answer>` 2 `</answer>` .

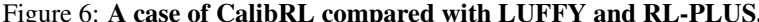

Figure 6: **A case of CalibRL compared with LUFFY and RL-PLUS.**

