# OpenReview forum: "Controllable Exploration in Hybrid-Policy RLVR for Multi-Modal Reasoning"
_ICLR.cc/2026/Conference — ICLR 2026 Poster_

### Official Review · Reviewer_ux1J · 2025-10-20

**Soundness:** 3
**Presentation:** 3
**Contribution:** 3
**Rating:** 6
**Confidence:** 4

**Summary:**

This paper addresses the challenge of entropy collapse and inefficient exploration in Reinforcement Learning with Verifiable Rewards (RLVR) for enhancing multi-modal large language model (MLLM) reasoning. Existing methods often suffer from policy degradation or over-exploitation, while hybrid approaches combining RL with supervised fine-tuning (SFT) can lead to distributional mismatch and accelerated entropy collapse. To overcome these issues, the authors propose CalibRL, a hybrid-policy RLVR framework that enables controllable exploration by treating expert demonstrations as a calibration baseline rather than a strict imitation target. CalibRL employs two key mechanisms: distribution-aware advantage weighting, which emphasizes rare yet informative responses to preserve exploration diversity , and an asymmetric LeakyReLU activation function that uses expert log-probabilities to moderate overconfident updates while amplifying underrepresented correct reasoning paths. By calibrating policy updates relative to expert knowledge, CalibRL aims to maintain productive policy entropy and guide exploration effectively, balancing exploration and exploitation. Extensive experiments on eight benchmarks demonstrate that CalibRL consistently outperforms standard GRPO and state-of-the-art hybrid-policy baselines in both in-domain and out-of-domain settings.

**Strengths:**

1. The paper addresses the crucial challenge of balancing exploration and exploitation within Reinforcement Learning with Verifiable Rewards (RLVR), a key paradigm for training reasoning capabilities in multi-modal large language models (MLLMs) which often suffers from entropy collapse.
2. The proposed CalibRL framework introduces intuitive mechanisms, namely distribution-aware advantage weighting and asymmetric activation using LeakyReLU calibrated by expert knowledge, which are conceptually straightforward to understand.
3. The effectiveness and generalization ability of CalibRL are rigorously validated through extensive experiments across eight diverse benchmarks, encompassing both in-domain and out-of-domain (OOD) reasoning tasks.

**Weaknesses:**

1. The comparison is limited to GRPO and a few hybrid methods (LUFFY, RL-PLUS), omitting recent advancements in GRPO variants like DAPO which could offer a more competitive benchmark .
2. While supplementary experiments are conducted on two additional models, the primary evaluation and analysis rely heavily on a single MLLM architecture (Qwen2.5-VL-7B), leaving the method's broad applicability across diverse MLLMs less certain .
3. The paper does not quantify the potential additional computational overhead introduced by calculating the log-probability gap relative to expert responses and applying the asymmetric activation and advantage weighting compared to the standard GRPO baseline.

**Questions:**

Please see paper weaknesses.

---

> ### Author Response · Authors · 2025-11-22
> **Response to Reviewer ux1J**
>
> Thank you for the time and effort invested in reviewing our work. Your positive comments and concrete suggestions are highly appreciated. We are encouraged by your feedback. Detailed responses to your questions are listed as follows.
>
> ## W1: Comparison with GRPO variants (e.g., DAPO).
> ---
>
> Thank you for this comment. We have validated the DAPO performance on the reported benchmark from our original manuscript. Results are shown in Table 14. As we can see, DAPO does not outperform GRPO in our setting. One possible explanation is that the higher clipping threshold leads to uncontrolled exploration, which further underscores the importance of our work on controllable exploration. These results have been updated to the revised manuscript.
>
> **Table 14: Performance comparison of GRPO, DAPO, and CalibRL. We present the Science benchmark as 'Sci.' and the Spatial Reasoning benchmark as 'Sp.'.**
>
> | Setting | GeoEval (in-domain) | Geo3K (in-domain) | GeoQA (in-domain) | Sci. (out-of-domain) | Sp. (out-of-domain) | MathVerse (out-of-domain) | MathVision (out-of-domain) | MathVista (out-of-domain) |
> |---------|---------------------|-------------------|-------------------|---------------------|---------------------|---------------------------|---------------------------|--------------------------|
> | GRPO | 26.15 | 39.77 | 52.52 | 61.99 | 48.02 | 49.01 | 27.89 | 70.00 |
> | DAPO | 25.19 | 40.93 | 52.52 | 61.61 | 50.21 | 47.82 | 27.07 | 70.40 |
> | CalibRL | 33.44 | 40.60 | 60.74 | 65.12 | 53.64 | 51.35 | 27.93 | 71.90 |
>
> ## W2: Extension evaluation across various MLLM architectures.
> ---
>
> Thank you for raising this important concern. Following your suggestion, we have expanded our evaluation by adding out-of-domain benchmarks on Qwen2.5-VL-3B and InternVL3-8B. Results are reported in Table 15. These results further demonstrate the broad applicability of our method across diverse MLLM architectures. We hope these additional experiments adequately address the concern.
>
> **Table 15: Expand performance comparison on different base models.**
>
> | Method | GeoEval | Geo3K | GeoQA | MathVision | MathVista |
> |--------|---------|-------|-------|------------|-----------|
> | **Qwen2.5VL-3B** |
> | GRPO | 15.11 | 28.95 | 40.05 | 22.99 | 65.1 |
> | LUFFY | 12.54 | 28.95 | 33.55 | 23.42 | 62.9 |
> | RL-PLUS | 12.00 | 27.79 | 35.15 | 22.99 | 63.4 |
> | CalibRL | 19.08 | 30.28 | 46.15 | 24.16 | 65.8 |
> | **IntrenVL3-8B** |
> | GRPO | 29.47 | 45.09 | 57.43 | 29.07 | 65.8 |
> | LUFFY | 13.83 | 15.14 | 48.67 | 29.07 | 62.0 |
> | RL-PLUS | 19.61 | 39.27 | 41.38 | 29.44 | 66.8 |
> | CalibRL | 31.08 | 48.59 | 58.89 | 30.06 | 68.5 |
>
> ## W3: Quantitative reporting of computational overhead.
> ---
>
> Thank you for this detailed question. The additional algorithmic components: computing the log-probability gap, applying the asymmetric LeakyReLU activation, and performing advantage weighting, are all simple element-wise operations with $O(n)$ complexity, where $n$ depends on the batch size and sequence length. Importantly, these operations do not require any forward or backward propagation and therefore do not dominate the computational cost; their overhead is negligible.
>
> Our method adds one off-policy expert response per prompt to the standard GRPO group of G policy-generated responses (where G = 10 in our implementation). This introduces a well-defined incremental cost that we quantify below (see Table 16). The expert response requires one additional forward pass to compute the expert log probability. Importantly, different from previous hybrid-policy methods, including LUFFY and RL-PLUS, the expert response **does not participate in gradient computation**. It serves only as a reference baseline. This means while we have G+1 forward passes per prompt, we still only perform G backward passes, **identical to standard GRPO**. Also, different from LUFFY, we require no additional reward for the expert trajectory, since we only include on-policy samples in the advantage calculation. Additionally, expert data collection is a one-time offline cost, not incurred during training, so rollout sampling overhead is zero.
>
> **Table 16: Per-prompt computational breakdown for a typical group size of G.**
>
> | Component | GRPO | CalibRL | Additional Cost |
> |-----------|------|---------|-----------------|
> | Rollout sampling | G | G | 0 |
> | Forward passes | G | G + 1 | +1 |
> | Reward model queries | G | G | 0 |
> | Backward passes | G | G | 0 |
>
> Total training time is nearly identical to GRPO. Rollout generation dominates iteration time and is unchanged. The additional forward pass adds overhead only to the training step, resulting in approximately 3% total wall-clock overhead for G. Despite this minimal overhead (one additional forward pass per prompt), our method achieves significant accuracy improvements over GRPO with the same on-policy rollout budget, demonstrating superior sample efficiency.

---

> > ### Comment · Reviewer_ux1J · 2025-11-24
> >
> > Thanks for your responses. My major concerns are addressed, and I would like to raise my score to accept. The authors are strongly recommended to include the extended experiments on DAPO and extra MLLMs into their final version of paper (if accepted).

---

> > > ### Author Response · Authors · 2025-11-24
> > > **Response to Reviewer ux1J**
> > >
> > > We are glad to hear that our responses have satisfactorily addressed your concerns. We have ensured that the extended experiments on DAPO and additional MLLMs are fully integrated into the revised manuscript. Thank you again for your effort and supportive feedback.

---

### Official Review · Reviewer_EZpA · 2025-10-31

**Soundness:** 3
**Presentation:** 3
**Contribution:** 2
**Rating:** 4
**Confidence:** 3

**Summary:**

The paper proposes **CalibRL**, a hybrid-policy RLVR framework for MLLM reasoning that reformulates expert supervision as a distributional baseline rather than a hard imitation target. For each prompt, the method defines a log-likelihood gap $\Delta \mathcal{l}$ between the model’s on-policy trajectory and an expert trajectory, multiplies it by a correctness indicator $s_i$, and **passes $−s_i\Delta \mathcal{l}$ through a LeakyReLU gate**. The resulting exploration term is further scaled by the **group-normalized advantage magnitude $∣\hat{A}_i∣$** (interpreted as group-wise rarity), and added to a GRPO-style objective. Intuitively, CalibRL **reinforces underweighted correct** responses and **suppresses overconfident incorrect** ones while keeping policy entropy from collapsing. Across eight benchmarks (in-domain and OOD) and multiple backbones, the method reports consistent gains over GRPO and several hybrid-policy baselines, with ablations on $\alpha$ (LeakyReLU slope) and $\lambda$ (trade-off).

**Strengths:**

i. **Clear and practical idea:** Uses the expert as a relative reference to guide exploration instead of enforcing imitation; integrates into GRPO with minimal code changes.

ii. **Readable and structured:** Algorithm and ablation setups are clear and reproducible.

iii. **Well-motivated gating:** The four-quadrant behavior (boost underconfident correct, damp overconfident wrong) neatly matches RL intuition and helps preserve entropy.

iv. **Consistent empirical gains** across diverse reasoning benchmarks and model sizes; ablations on $\alpha$ and $\lambda$ illustrate controllability of exploration.

v. **Rare-event emphasis** via $∣\hat{A}_i∣$ often improves signal quality in sparse-reward, group-sampled settings.

**Weaknesses:**

i. **$∣\hat{A}_i∣$ is under-specified** near Eq. (9): the paper states it “captures group-wise rarity” but does not restate the exact computation (presumably GRPO’s group z-score) nor the degenerate case handling when the group reward variance is zero (all-correct or all-wrong).

ii. **Length bias in $\Delta \mathcal{l}$:** using sequence-level log-likelihood differences without explicit length normalization can penalize longer (yet correct) solutions; no targeted analysis is provided.

iii. **Sensitivity/fairness:** Results depend on group size $G$, sampling temperature/top-p, and the presence/absence of KL to a reference policy. These choices for baselines (e.g., LUFFY, RL-PLUS) are not exhaustively justified or tuned for parity; statistical significance over multiple seeds is limited.

iv. **Potential double-counting of advantage:** GRPO already weights updates by $∣\hat{A}_i∣$,$t$; the exploration term scales again by $∣\hat{A}_i∣$. Without gradient-norm breakdowns, it’s unclear whether this over-amplifies certain samples.

v. **Theory is light:** The claim that $∣\hat{A}_i∣$ encodes “rarity” is intuitive but lacks formal characterization (e.g., relation to group frequency, robustness to reward noise).

**Questions:**

1. Please **explicitly define $∣\hat{A}_i∣$** used in Eq. (9): is it exactly the GRPO group z-score at sequence level (shared across tokens)? Do you apply clipping or normalization? How do you handle **std = 0** groups (all-correct/all-wrong)?

2. How do you address **length effects** in $\Delta \mathcal{l}$? Any experiments with token-average log-prob or length-normalized variants, and their impact on long CoT tasks?

3. Does CalibRL remain effective if the **expert baseline is weaker** (e.g., a smaller SFT model)? Please report performance vs. strength of the expert.

---

> ### Author Response · Authors · 2025-11-22
> **Response to Reviewer EZpA Part 1/3**
>
> We sincerely thank you for your thoughtful review. Your positive comments on our ideas and results are greatly appreciated. We also value your suggestions, which highlight important directions for improvement. We address your main concerns as follows.
>
> ## W1: Further Explanation and Clarification of $|\hat{A}_i|$ in Eq. (9).
> ---
>
> ### Explicit definition of $|\hat{A}_i|$
>
> Thank you for pointing this out and for providing us with the opportunity to clarify our presentation. Your interpretation of $|\hat{A}_i|$ is right: the $|\hat{A}_i|$ denotes the absolute value of the group-wise advantage. We use the same symbol as Eq. (3) for consistency. In our implementation, we omit the standard-error normalization (std) from the original GRPO, in line with the Dr.GRPO implementation. This design choice is consistent with the LUFFY and RL-PLUS settings described in Appendix B.
>
> We compute this group advantage as the deviation from the group mean and then take its absolute value:
>
> $$
> \hat{A}\_{i,t} = R(\tau_i)-\textit{mean}(R(\{\tau_i\}^G\_{i=1})), |\hat{A}\_{i,t}| = |R(\tau_i)-\textit{mean}(R(\{\tau_i\}^G\_{i=1}))|.
> $$
>
> ### 'Zero-advantages' cases
>
> We appreciate your careful observation. The issue of "zero advantage" when a group contains all-correct or all-wrong answers is indeed a well-recognized challenge in GRPO improvements. LUFFY, when exploring the boundaries of on-policy capabilities, introduced the "mix-policy" strategy. This framework naturally incorporates a correct expert sample in the calculation of group advantages, which inherently resolves the zero-advantage problem when the model answers all incorrectly. Several methods, such as TemplateRL, have been inspired by LUFFY to investigate this issue further.
>
> Our approach, however, focuses more on controllable exploration within such a hybrid-policy framework. Similar to RL-PLUS, we only include on-policy samples in the advantage calculation, while expert trajectories are used for potential corrections. We do not apply special handling for the zero-advantage case. When all group rewards are identical, $|\hat{A}_i|=0$ for every sample, meaning both the policy-gradient term in Eq. (10) and our exploration loss evaluate to zero, and no gradient is contributed by that group.
>
> Empirical results indicate that the zero-advantage issue does not significantly affect the effectiveness of our approach. We are glad that the reviewer shares our concern about this issue, and we look forward to further addressing it in future work, especially in combination with our controllable exploration research.
>
> ## W2: Analysis of potential length bias in $\Delta \ell_i$.
> ---
>
> Thank you for the careful observation. We acknowledge that using the $\Delta \ell_i$ may introduce a mild length preference. However, in our method, the expert trajectory is intended to guide not only correctness but also the answering paradigm—including stylistic preferences such as avoiding overly long or excessively short responses. In this sense, a small length preference is aligned with our design goal of encouraging the model to follow the expert's response style.
>
> Moreover, this bias does not override correctness learning. The policy is still optimized primarily by the main GRPO loss, which determines reward-maximizing behavior. The exploration term is scaled by a small $\lambda$ and serves only to regulate the degree of deviation from the expert, rather than to dictate correctness. Thus, the optimization direction remains dominated by the policy objective, and we did not observe harmful interference with correctness. We appreciate this observation and have added clarification in the revised manuscript.
>
> To further analyze the influence of such length bias, we conduct an ablation by adding a length normalization to the $\Delta \ell_i$ computation. Results are reported in Table 12. The strong performance of the model without length normalization supports our original design.
>
> **Table 12: Performance on length norm.**
>
> | Method | GeoEval (in-distribution) | Geo3K (out-of-distribution) | GeoQA (out-of-distribution) | Avg. |
> |--------|---------------------------|----------------------------|----------------------------|------|
> | CalibRL w/o length norm | 33.44 | 40.60 | 60.74 | 44.93 |
> | CalibRL w/ length norm | 29.26 | 39.10 | 60.34 | 42.90 |

---

> ### Author Response · Authors · 2025-11-22
> **Response to Reviewer EZpA Part 2/3**
>
> ## W3: Clarification on baseline setting for LUFFY and RL-PLUS.
> ---
>
> Thank you for the thoughtful comment. We acknowledge the validity of this concern. The training configurations, including sampling temperature, top-p, and the absence of KL regularization to a reference policy, are **directly inherited from the compared baselines (LUFFY and RL-PLUS)**. We retain these settings for our implementation of the baselines to ensure comparability. For our training, we keep all controllable hyperparameters (e.g., group size G) consistent across experiments, while using the publicly released default parameters for each baseline, thereby ensuring a fair and faithful comparison. Details are described in Appendix B.
>
> ## W4: Clarification on possible double-counting of advantage terms.
> ---
>
> Thank you for bringing this important point to our attention. We would like to clarify that our method does not double-count the advantage nor amplify gradients undesirably. Our training objective:
>
> $$
> \mathcal{J}(\theta) =
> \mathbb{E}\_{q \sim \mathcal{D}, \tau \sim \pi\_\theta(\cdot|q)}\Bigg[
>     \sum_{t=1}^{|\tau|} \min\Big(
>         r\_{i,t}(\theta)\hat{A}\_{i,t}, \\
>     \quad \text{clip}\big(
>         r\_{i,t}(\theta), 1-\epsilon, 1+\epsilon\big)\hat{A}\_{i,t}
>     \Big) - \lambda |\hat{A}_i| \cdot \text{LeakyReLU}\big(-s_i \Delta \ell_i, \alpha\big)
> \Bigg].
> $$
>
> In the main GRPO objective, $\hat{A}_{i,t}$ determines the update direction of the policy, while in the exploration term, $|\hat{A}_i|$ is treated purely as a static importance weight. The two occurrences of the advantage influence different aspects of the optimization: the GRPO term governs policy improvement, whereas the exploration term modulates the strength of trajectory-level exploration correction.
>
> The exploration loss gradient is:
>
> $$
> \frac{\partial \mathcal{L}_{\mathrm{exploration}}}{\partial \theta} = |\hat{A}_i| \cdot \text{LeakyReLU}'(-s_i\Delta\ell_i) \cdot (-s_i) \cdot \nabla\_\theta\log\pi\_\theta(a_i),
> $$
>
> while the GRPO gradient is:
>
> $$
> \frac{\partial}{\partial \theta}\big(r_{i,t}(\theta)\hat{A}_{i,t}\big)
> = \hat{A}\_{i,t} \cdot \nabla\_\theta r\_{i,t}(\theta) = \hat{A}\_{i,t}
> \cdot r\_{i,t}(\theta) \cdot \nabla\_\theta\log\pi\_\theta(a\_{i,t}).
> $$
>
> Both terms produce updates through $\nabla_\theta\log\pi_\theta$, but with different multiplicative coefficients (the GRPO term uses $\hat{A}_{i,t}r_{i,t}$ while the exploration term uses $|\hat{A}_i|(-s_i\mathrm{LeakyReLU}(\cdot))$ and is scaled by $\lambda$). The key observation is that these coefficients are additive in the total gradient, not multiplicative. The final gradient is:
>
> $$
> \nabla\_\theta \mathcal{J} = \sum_t \hat{A}\_{i,t} \cdot r\_{i,t}(\theta) \cdot \nabla\_\theta\log\pi_\theta(a\_{i,t}) - \lambda \cdot |\hat{A}_i| \cdot (-s_i) \cdot \text{LeakyReLU}(\cdot) \cdot \nabla\_\theta\log\pi\_\theta(a_i).
> $$
>
> The advantage appears in two separate, additive gradient contributions that can constructively or destructively interfere depending on their signs, with the hyperparameter $\lambda$ controlling the relative magnitude of exploration correction versus policy improvement, preventing uncontrolled amplification. In our experiments, $\lambda = 0.1$ ensures the exploration term provides a measured correction signal without dominating the GRPO updates.
>
> We hope this clarifies the concern, and we sincerely appreciate your insightful question.
>
> ## W5: Further theoretical grounding of the "rarity" interpretation.
> ---
>
> Thank you for recognizing the intuition of using $|\hat{A}_i|$ value to naturally capture group-wise "rarity". We now provide a more formal characterization of it.
>
> ### Relation to group frequency
>
> First, consider several example groups of sequence-level rewards:
>
> - **Group 1**: [0,1,1,1], where the "0" answer is relatively rare within the group. The absolute value of the group advantage is [0.75,0.25,0.25,0.25]. The "0" has the largest $|\hat{A}|$, indicating that a reward of "0" is the group-wise outlier.
> - **Group 2**: [1,0,0,0], where the "1" answer is relatively rare within the group. This group resulted in the same absolute value of the group advantage: [0.75,0.25,0.25,0.25], also indicating that a reward of "1" is "rare" in this group.
>
> The examples demonstrate that the frequency of correct/incorrect answers and $|\hat{A}_i|$ correspond symmetrically and automatically, without requiring additional heuristics. We further show a continuous change between $|\hat{A}_i|$ and reward frequency in a group of 10 samples. **Please refer to Figure 4 in the revised manuscript on Page 18 (Appandix E)** The curve shows a strictly monotonic mapping between rarity and magnitude.

---

> ### Author Response · Authors · 2025-11-22
> **Response to Reviewer EZpA Part 3/3**
>
> ### Robustness to small reward noise
>
> To analyze robustness under small additive reward noise (e.g., formatting bonuses), consider the perturbed reward
>
> $$
> R'(\tau_i) = R(\tau_i) + \delta_i,\qquad |\delta_i|\le \epsilon.
> $$
>
> The corresponding advantage becomes
>
> $$
> \hat{A}'\_{i,t}
> = R'(\tau_i) - \mathrm{mean}(R'(\{\tau_j\}\_{j=1}^G))
> = (R(\tau_i)-\mu_G) + (\delta_i - \bar{\delta}),
> $$
>
> where
>
> $$
> \mu_G = \mathrm{mean}(R(\{\tau_j\}\_{j=1}^G)),\qquad
> \bar{\delta} = \mathrm{mean}(\{\delta_j\}\_{j=1}^G).
> $$
>
> Thus, the perturbation to the advantage is
>
> $$
> \hat{A}'\_{i,t} - \hat{A}\_{i,t} = \delta_i - \bar{\delta},
> $$
>
> which is a mean-centered noise term satisfying
>
> $$
> |\delta_i - \bar{\delta}| \le 2\epsilon.
> $$
>
> Since typically
>
> $$
> |R(\tau_i)-\mu_G| = O(1),
> $$
>
> the ordering of $|\hat{A}_{i,t}|$, which determines group-wise rarity, is preserved for sufficiently small $\epsilon$. In the common case where noise is approximately uniform across samples (e.g., shared formatting bonuses), we have $\delta_i \approx \bar{\delta}$, and the perturbation cancels:
>
> $$
> \hat{A}'\_{i,t} \approx \hat{A}\_{i,t}.
> $$
>
> Hence, mean-centered normalization makes the rarity signal $|\hat{A}_{i,t}|$ inherently robust to small reward noise.
>
> We appreciate the opportunity to strengthen our presentation and have incorporated the above discussion into the revised manuscript.
>
> ## Q1: Detailed definition of the $|\hat{A}_i|$ in Eq. (9). And how to handle 'zero-advantages cases'.
> ---
>
> We hope our response to **W1** clarifies the issue and addresses your concern.
>
> ## Q2: Additional result to address length effects.
> ---
>
> We conduct an ablation by adding a length normalization to the $\Delta \ell_i$ computation. Results are reported in Table 12. The strong performance of the model without length normalization supports our original design. We hope our response to **W2** clarifies the issue and addresses your concern.
>
> ## Q3: Effect of weaker expert baselines.
> ---
>
> Thank you for this comment. In our original manuscript, we use GPT-4o as the expert to generate CoT responses for training. To further verify the generalizability of CalibRL, we additionally compare GRPO and CalibRL when the training data are produced by Qwen2.5-VL-72B, which serves as a weaker expert compared to GPT-4o. We keep the size of the Qwen-generated dataset comparable to the GPT-generated one (approximately 9k samples). However, because the two expert models differ in capability, the filtered sets of correct responses are not identical, and we cannot guarantee that both methods are trained on the same sampled data. Nonetheless, this setting still provides a meaningful evaluation of the robustness and generality of CalibRL under varying expert quality.
>
> The results in Table 13 show that CalibRL consistently delivers substantial improvements over GRPO in both settings. As expected, the magnitude of improvement depends on the quality of the expert baseline, with a stronger expert providing a high-quality,  generalizable, and informative reference. This highlights the importance of expert guidance in controllable-entropy RLVR and further demonstrates that CalibRL can effectively adapt its learning behavior to the quality of the expert model, leading to significant gains in training performance.
>
> **Table 13: Trained on different expert data.**
>
> | Method | GeoEval (in-distribution) | Geo3K (out-of-distribution) | GeoQA (out-of-distribution) | Avg. |
> |--------|---------------------------|----------------------------|----------------------------|------|
> | **On Qwen data** |
> | GRPO | 23.26 | 41.10 | 49.87 | 38.08 |
> | CalibRL | 26.90 | 40.43 | 56.90 | 41.41 (+3.33) |
> | **On GPT data** |
> | GRPO | 26.15 | 39.77 | 52.52 | 39.48 |
> | CalibRL | 33.44 | 40.60 | 60.74 | 44.93 (+5.45) |

---

> > ### Comment · Reviewer_EZpA · 2025-11-25
> >
> > Thank you for the response and the additional details. The most of my concerns are resolved. I raise the score.

---

> > > ### Author Response · Authors · 2025-11-26
> > > **Response to Reviewer EZpA**
> > >
> > > We sincerely appreciate your experimental and theoretical advice to improve the quality of the manuscript. We are also very glad to see that we have addressed your concerns. Thank you for your positive support. Please let us know if you have further advice and suggestions.

---

### Official Review · Reviewer_5pSB · 2025-11-01

**Soundness:** 3
**Presentation:** 3
**Contribution:** 3
**Rating:** 6
**Confidence:** 2

**Summary:**

I am not an expert in this domain, I will try to review this paper from my limited understanding.

This work introduces CalibRL, a reinforcement learning framework that improves exploration in multi-modal large language models. Unlike prior RLVR methods prone to entropy collapse, CalibRL uses advantage weighting and a LeakyReLU-based asymmetric activation to guide exploration with expert supervision while preserving diversity. This balanced approach stabilizes training and enhances reasoning performance, achieving consistent gains over GRPO, LUFFY, and RL-PLUS across multiple benchmarks.

**Strengths:**

1. The paper is clearly written and easy to follow.
2. It addresses an important and widely existing problem, that under the SFT-then-RL paradigm, the policy becomes tightly anchored to the expert distribution during the SFT stage. This causes exploration to be restricted within the local neighborhood of expert behaviors, making it difficult to adapt to reward signals or discover more optimal reasoning trajectories.
3. The paper provides comprehensive and convincing ablation studies that support its claims.

**Weaknesses:**

1. Most evaluations are math benchmarks. The claim of general multi-modal reasoning would be more convincing with benchmarks involving richer visual, linguistic, or commonsense reasoning modalities (e.g., ScienceQA, MMMU, or multimodal dialogue tasks).

**Questions:**

1. The proposed method does not seem inherently limited to VLMs. It could also apply to LLMs. Why is the paper specifically focused on VLMs?
2. Why LeakyReLU-based activation yields controllable entropy growth? Do other similar activations have a similar effect?

---

> ### Author Response · Authors · 2025-11-22
> **Response to Reviewer 5pSB Part 1/2**
>
> We are truly grateful for your time and effort invested in reviewing our work. Your positive feedback has been both encouraging and instrumental in further improving our paper. Herein, we address the points raised in your comments.
>
> ## W1: Broadening evaluation on commonsense reasoning benchmarks.
> ---
>
> Thank you for this comment. In Table 2 of the original manuscript, we have included the out-of-domain results, which also involve science problems and spatial reasoning tasks. We further validated our best-trained model on the suggested MMMU **[5]** and ScienceQA **[6]**. The new results are reported in Table 9 and have been added to the revised manuscript.
>
> Notably, the core purpose of MMMU and ScienceQA is to measure the breadth of a model’s knowledge. MMMU spans 6 academic fields, 30 college-level subjects, and 183 sub-domains. ScienceQA consists of 21,208 science questions (from elementary to high school curricula) covering multiple scientific disciplines. On these benchmarks, CalibRL outperforms other RL methods. We attribute this to the generalizable reasoning patterns learned by CalibRL. However, we acknowledge that the performance gains here are more modest compared to multimodal reasoning tasks. This is because RL-based post-training has limited capacity to expand the inherent knowledge boundaries of the base model.
>
> **Table 9: Performance on commonsense reasoning benchmarks.**
>
> | Method | MMMU | ScienceQA |
> |--------|------|-----------|
> | grpo | 55.44 | 88.34 |
> | luffy | 55.22 | 87.95 |
> | rlplus | 55.88 | 87.5 |
> | CalibRL | 56.55 | 89.04 |
>
> ## Q1: Clarification on the focus on VLMs versus LLMs.
> ---
>
> Thank you for this comment. Our method can certainly be expanded to broader applications beyond visual reasoning.
>
> In this work, we focus on achieving stable entropy control in RLVR for VLMs, where the challenge becomes especially severe in the vast state space of MLLMs and substantially limits learning efficiency. While a series of works like LUFFY **[1]** and RL-PLUS **[2]** have achieved promising results for LLMs, such results have not transferred to MLLMs, leaving a notable gap in the literature, one that our work aims to fill.
>
> We conduct an experiment to validate the effectiveness of our CalibRL beyond visual reasoning. We train Qwen2.5-VL-7B on a 9k-sample subset of the MATH dataset **[3]** and evaluate the model on the in-distribution benchmark MATH-500 **[3]** and the out-of-distribution benchmark AMC **[4]**.
>
> Results are reported in Table 10. Training on pure text data, CalibRL yields stronger gains than GRPO on MATH-500, demonstrating more effective in-distribution improvement. Moreover, unlike GRPO, which fails to enhance out-of-distribution performance, CalibRL continues to deliver benefits on AMC, achieving strong generalization results. It is worth noting that due to time constraints, we did not use large-scale training data (e.g., 45k) as in the math reasoning methods  LUFFY **[1]**  and RL-PLUS **[2]**, nor did we tune the hyperparameters specifically for pure text tasks; instead, we reused the empirical hyperparameters from multi-modal reasoning.
>
> **Table 10: Performance of the Qwen2.5-VL-7B model on math reasoning tasks.**
>
> | Method | MATH-500 (in-distribution) | AMC (out-of-distribution) |
> |--------|---------------------------|---------------------------|
> | Qwen2.5-VL-7B | 64.8 | 31.97 |
> | GRPO | 68.5 | 31.59 |
> | CalibRL | 70.2 | 32.08 |

---

> ### Author Response · Authors · 2025-11-22
> **Response to Reviewer 5pSB Part 2/2**
>
> ## Q2: Comparing LeakyReLU with similar activations and its controllable entropy behavior.
> ---
>
> Thank you for this useful comment. Adopting the leaky mechanism of the LeakyReLU function enables us to build a controllable entropy–shaping mechanism, facilitated by its adjustable negative slope. We show the necessity of this design through ablations using other activation functions, where the ReLU, sigmoid, and Huber fully truncate negative values, provide no controllable scaling, and the tanh, which cannot modulate the magnitude, none of which offer the required level of control. We train the Qwen2.5-VL-7B, using the same training data we constructed for our main paper results. Results are reported in Table 4, the three ReLU, sigmoid, and Huber either failed to improve the GRPO baseline or only provided a slight gain. On the other hand, the tanh provides a relatively strong improvement, showing the importance of the negative value in the entropy-shaping mechanism. Finally, our design for the CalibRL consistently achieved the highest performance.
>
> **Table 11: Ablations on activation functions.**
>
> | Method | GeoEval (in-distribution) | Geo3K (out-of-distribution) | GeoQA (out-of-distribution) | Avg. |
> |--------|---------------------------|----------------------------|----------------------------|------|
> | GRPO | 26.15 | 39.77 | 52.52 | 39.48 |
> | CalibRL w/ ReLU | 26.47 | 33.11 | 57.96 | 39.18 |
> | CalibRL w/ Sigmoid | 26.69 | 32.61 | 60.08 | 39.79 |
> | CalibRL w/ Huber | 24.22 | 27.62 | 55.70 | 35.85 |
> | CalibRL w/ Tanh | 30.65 | 40.93 | 58.62 | 43.40 |
> | CalibRL w/ LeakyReLU($\alpha=0.5$) | **33.44** | **40.60** | **60.74** | **44.93** |
>
> ## References
> ---
>
> **[1]** Jianhao Yan, Yafu Li, Zican Hu, Zhi Wang, Ganqu Cui, Xiaoye Qu, Yu Cheng, and Yue Zhang. Learning to reason under off-policy guidance. *arXiv preprint arXiv:2504.14945*, 2025.
>
> **[2]** Yihong Dong, Xue Jiang, Yongding Tao, Huanyu Liu, Kechi Zhang, Lili Mou, Rongyu Cao, Yingwei Ma, Jue Chen, Binhua Li, et al. Rl-plus: Countering capability boundary collapse of llms in reinforcement learning with hybrid-policy optimization. *arXiv preprint arXiv:2508.00222*, 2025.
>
> **[3]** Dan Hendrycks, Collin Burns, Saurav Kadavath, Akul Arora, Steven Basart, Eric Tang, Dawn Song, and Jacob Steinhardt. Measuring mathematical problem solving with the math dataset. *arXiv preprint arXiv:2103.03874*, 2021.
>
> **[4]** Jia Li, Edward Beeching, Lewis Tunstall, Ben Lipkin, Roman Soletskyi, Shengyi Huang, Kashif Rasul, Longhui Yu, Albert Q Jiang, Ziju Shen, et al. Numinamath: The largest public dataset in ai4maths with 860k pairs of competition math problems and solutions. *Hugging Face repository*, 13(9):9, 2024.
>
> **[5]** Xiang Yue, Yuansheng Ni, Kai Zhang, Tianyu Zheng, Ruoqi Liu, Ge Zhang, Samuel Stevens, Dongfu Jiang, Weiming Ren, Yuxuan Sun, Cong Wei, Botao Yu, Ruibin Yuan, Renliang Sun, Ming Yin, Boyuan Zheng, Zhenzhu Yang, Yibo Liu, Wenhao Huang, Huan Sun, Yu Su, and
> Wenhu Chen. Mmmu: A massive multi-discipline multimodal understanding and reasoning benchmark for expert agi. In Proceedings of CVPR, 2024.
>
> **[6]** Pan Lu, Swaroop Mishra, Tony Xia, Liang Qiu, Kai-Wei Chang, Song-Chun Zhu, Oyvind Tafjord, Peter Clark, and Ashwin Kalyan. Learn to explain: Multimodal reasoning via thought chains for science question answering. In The 36th Conference on Neural Information Processing Systems (NeurIPS), 2022.

---

> > ### Comment · Reviewer_5pSB · 2025-11-27
> >
> > Thanks for the clarification, which solves my concerns. I will keep the score as it is.

---

> > > ### Author Response · Authors · 2025-11-28
> > > **Response to Reviewer 5pSB**
> > >
> > > We are very glad to hear that our responses have addressed your concerns. We sincerely appreciate your thoughtful advice and the time you have taken to help us improve our work. Thank you for your positive support. Please let us know if you have further advice and suggestions.

---

### Official Review · Reviewer_sgYA · 2025-11-03

**Soundness:** 3
**Presentation:** 3
**Contribution:** 2
**Rating:** 4
**Confidence:** 4

**Summary:**

CalibRL reframes demonstrations as a calibration baseline inside RL with verifiable rewards, adding two simple mechanisms to a GRPO pipeline: a relative log‑probability gap between the on‑policy and expert responses passed through an asymmetric (LeakyReLU‑style) gate, and a group‑rarity weighting that scales updates by how uncommon a response is within a rollout group. The result is exploration that stays stochastic yet directed—rare correct behaviors are strengthened while overconfident mistakes are damped. Across eight visual‑math and multimodal reasoning benchmarks and several base models, the method consistently improves over a GRPO baseline and avoids the entropy collapse or aimless randomness observed in common SFT‑then‑RL and prior hybrid setups.

**Strengths:**

1. This proposed method introduces a clear hybrid‑policy objective that treats expert trajectories as a relative reference, which helps maintain entropy while steering updates toward verified behaviors. The design uses an intuitive pair of knobs—an asymmetric gate on the policy‑vs‑expert log‑prob gap and a group‑rarity magnitude—that integrates cleanly with GRPO and is easy to implement.

2. Simple, general mechanism: The LeakyReLU‑gated is an elegant way to use demonstrations for relative guidance. It respects the on‑policy signal and works as a plug‑in to GRPO.

3. Experiments demonstrate consistent gains on eight benchmarks, with especially large improvements on the hardest geometry split and solid out‑of‑domain boosts. The approach generalizes across different model sizes/architectures, and the ablations probe the important controls (gate slope and balance weight) to show how exploration is tuned.

**Weaknesses:**

1. GeoEval split clarity: The paper constructs GeoEval from validation failures of GPT‑4o CoT filtering, then reports it as a test benchmark with the largest deltas. Please clarify whether this split was ever used for hyper-parameter tuning or early stopping. If yes, results could be optimistically biased; if no, state this explicitly and detail safeguards.

2. Baselines for entropy control: Since the contribution is controllable exploration, it misses comparisons to standard entropy‑regularized GRPO (fixed/annealed entropy bonus) or token‑level entropy‑regularized updates. This would isolate whether your calibrated, expert‑relative update is better than “just add entropy.”

3. Ablation on the activation & reference. It would strengthen claims to compare LeakyReLU vs hinge, Huber, or sigmoid saturations, and to test “expert baseline” vs “reference policy baseline” (e.g., KL to SFT policy), to confirm that expert pairing and asymmetry specifically drive the effect.

4. Compute/efficiency reporting. Please quantify incremental overhead (extra forward passes to score expert trajectories), wall‑clock time vs GRPO, and sample efficiency (accuracy vs total rollouts). This is particularly important for practitioners training on LLMs / VLMs.

**Questions:**

1. Why do LUFFY/RL‑PLUS trail GRPO here? Any hints from your logs (entropy trends, clipping ratios, off‑policy mismatch) as to why these hybrids underperform? Sharing failure modes would be valuable to the community.

2. Calibration evidence. Beyond entropy curves, could you report ECE or other calibration diagnostics on answer probabilities, to substantiate “distributional calibration”?

3. Generalization beyond geometry. Have you tried training CalibRL outside geometry (e.g., coding or algebraic word problems) with appropriate verifiers (e.g., test oracles for code)? It would help demonstrate domain generality.

---

> ### Author Response · Authors · 2025-11-22
> **Response to Reviewer sgYA Part 1/3**
>
> We are truly thankful for the thoughtful remarks and the experimental recommendations you have provided. These suggestions shed light on what we can improve, and we believe they will be instrumental in refining our work further. Below are our responses to your concerns.
>
> ## W1: GeoEval split clarity.
> ---
>
> Thank you for the thoughtful comment. We acknowledge the validity of this concern. The training data and the GeoEval split were constructed based on GPT-4o CoT filtering before any model training began. The GeoEval data remained **completely unseen during training**. We trained all models for a fixed number of steps without any early stopping, and checkpoints for comparison were selected at identical training steps across all experiments. As shown in our ablation studies (Section 4.3 on Page 8), hyper-parameter tuning was conducted by jointly evaluating performance across all benchmarks, rather than relying specifically on GeoEval. We appreciate the question and have clarified this accordingly in the revised manuscript.
>
> ## W2: Additional baselines for entropy control and comparison.
> ---
>
> Thank you for this valuable comment, which will help us improve the completeness of our validation. We conduct ablations to compare our controllable exploration with several entropy regularization methods. We train the Qwen2.5-VL-7B, using the same training data we constructed for our main paper results.
>
> Results are reported in Table 3. In summary, **only CalibRL delivers consistent improvements** across all evaluated benchmarks. We first compare our method with fixed-coefficient entropy regularization by setting the entropy coefficient to 0.01 according to **[1, 2]**, which results in a degradation in performance. We then apply the widely used entropy-control mechanisms based on KL and clip-covariance regularization **[5]** on top of GRPO. The KL-Cov variant provides a slight improvement on some tasks but remains noticeably weaker than our CalibRL, while the Clip-Cov variant again results in a performance drop. Compared with conventional entropy-based methods, CalibRL enhances policy entropy in a guided manner that avoids unguided randomness and directs exploration toward meaningful reasoning behavior. As a result,  it achieves more effective exploration than the compared entropy-based baselines.
>
> **Table 3: Performance comparison of different entropy control.**
>
> | Method | GeoEval (in-distribution) | Geo3K (out-of-distribution) | GeoQA (out-of-distribution) | Avg. |
> |--------|---------------------------|----------------------------|----------------------------|------|
> | GRPO | 26.15 | 39.77 | 52.52 | 39.48 |
> | + entropy coefficient = 0.01 | 22.62 | 40.27 | 47.48 | 36.79 |
> | + KL-Cov | 26.47 | 41.60 | 53.58 | 40.55 |
> | + Clip-Cov | 25.40 | 39.43 | 52.12 | 38.98 |
> | **CalibRL** | **33.44** | **40.60** | **60.74** | **44.93** |
>
> ## W3: Further ablations on the activation function and reference policy.
> ---
>
> ### Ablations on different activation functions
>
> Thank you for this useful comment. Adopting the leaky mechanism of the LeakyReLU function enables us to build a controllable entropy–shaping mechanism, facilitated by its adjustable negative slope. We show the necessity of this design through ablations using other activation functions, where the ReLU, sigmoid, and Huber fully truncate negative values, provide no controllable scaling, and the tanh, which cannot modulate the magnitude, none of which offer the required level of control. We train the Qwen2.5-VL-7B, using the same training data we constructed for our main paper results. Results are reported in Table 4, the three ReLU, sigmoid, and Huber either failed to improve the GRPO baseline or only provided a slight gain. On the other hand, the tanh provides a relatively strong improvement, showing the importance of the negative value in the entropy-shaping mechanism. Finally, our design for the CalibRL consistently achieved the highest performance.
>
> **Table 4: Ablations on activation functions.**
>
> | Method | GeoEval (in-distribution) | Geo3K (out-of-distribution) | GeoQA (out-of-distribution) | Avg. |
> |--------|---------------------------|----------------------------|----------------------------|------|
> | GRPO | 26.15 | 39.77 | 52.52 | 39.48 |
> | CalibRL w/ ReLU | 26.47 | 33.11 | 57.96 | 39.18 |
> | CalibRL w/ Sigmoid | 26.69 | 32.61 | 60.08 | 39.79 |
> | CalibRL w/ Huber | 24.22 | 27.62 | 55.70 | 35.85 |
> | CalibRL w/ Tanh | 30.65 | 40.93 | 58.62 | 43.40 |
> | CalibRL w/ LeakyReLU ($\alpha=0.5$) | **33.44** | **40.60** | **60.74** | **44.93** |

---

> ### Author Response · Authors · 2025-11-22
> **Response to Reviewer sgYA Part 2/3**
>
> ### Ablations on different reference policies
>
> Thank you for this useful comment. We conduct ablations to compare different reference policies for computing $\Delta \ell_i$. Specifically, we replace the $\log \pi_\theta(\tau_i^{\text{expert}} | q_i)$ with the reference policy $ \log \pi_\theta(\tau_i^{\text{ref}} | q_i)$, resulting in a log-probability gap as:
>
> $$
> \Delta \ell_i' = \log \pi_\theta(\tau_i^{\text{policy}} | q_i) - \log \pi_\theta(\tau_i^{\text{ref}} | q_i)
> $$
>
> We train the Qwen2.5-VL-7B, using the same training data we constructed for our main paper results. Results are reported in Table 5. The expert baseline strongly suppresses the reference policy baseline, showing the importance of the expert guidance in our controllable exploration design.
>
> **Table 5: Ablations on reference baselines.**
>
> | Method | GeoEval (in-distribution) | Geo3K (out-of-distribution) | GeoQA (out-of-distribution) | Avg. |
> |--------|---------------------------|----------------------------|----------------------------|------|
> | CalibRL w/ expert baseline | 33.44 | 40.60 | 60.74 | 44.93 |
> | CalibRL w/ ref policy baseline | 27.87 | 40.77 | 55.57 | 42.74 |
>
> ## W4: Reporting of computational and sample efficiency.
> ---
>
> Thank you for bringing this important practical concern to attention. Our method adds one off-policy expert response per prompt to the standard GRPO group of G policy-generated responses (where G = 10 in our implementation). This introduces a well-defined incremental cost that we quantify below (see Table 6). The expert response requires one additional forward pass to compute the expert log probability. Importantly, different from previous hybrid-policy methods, including LUFFY and RL-PLUS, the expert response **does not participate in gradient computation**. It serves only as a reference baseline. This means while we have G+1 forward passes per prompt, we still only perform G backward passes, **identical to standard GRPO**. Also, different from LUFFY, we require no additional reward for the expert trajectory, since we only include on-policy samples in the advantage calculation. Additionally, expert data collection is a one-time offline cost, not incurred during training, so rollout sampling overhead is zero.
>
> **Table 6: Per-prompt computational breakdown for a typical group size of G.**
>
> | Component | GRPO | CalibRL | Additional Cost |
> |-----------|------|---------|-----------------|
> | Rollout sampling | G | G | 0 |
> | Forward passes | G | G + 1 | +1 |
> | Reward model queries | G | G | 0 |
> | Backward passes | G | G | 0 |
>
> Total training time is nearly identical to GRPO. Rollout generation dominates iteration time and is unchanged. The additional forward pass adds overhead only to the training step, resulting in approximately 3% total wall-clock overhead for G. Despite this minimal overhead (one additional forward pass per prompt), our method achieves significant accuracy improvements over GRPO with the same on-policy rollout budget, demonstrating superior sample efficiency.
>
> ## Q1: Insights into why LUFFY and RL-PLUS underperform.
> ---
>
> Thank you for this insightful question. We obtain several statistical data points from our trained checkpoints to reveal a shared structural failure mode across LUFFY and RL-PLUS. Both methods attempt to use experts without an effective calibration mechanism. This leads to either diluted expert signals or unstable confidence dynamics.
>
> LUFFY collapses to a single expert mode. The policy and expert distributions become nearly indistinguishable after training. The model treats expert responses as a uniform style to imitate. It loses the ability to evaluate where expert guidance should matter and where exploration should continue. RL-PLUS shows the opposite tendency. The policy becomes prematurely certain. Expert responses also lose diversity and drift toward the policy. The model reinforces its own early preferences without checking them against a stable expert reference. It converges quickly but without reliable calibration. In both cases, expert information does not form a stable reference for exploration.
>
> CalibRL addresses this missing mechanism. It does not aim to imitate expert answers. It treats experts as a reference point that calibrates the policy's search direction. The model explores actively while remaining aware of expert quality. This calibrated exploration avoids problems of LUFFY and RL-PLUS.
>
> **Table 7: Statistics.**
>
> | Method | $\Delta \ell_i$ | policy entropy | reward | response length (expert length 382.19) |
> |--------|------|----------------|--------|-----------------|
> | Qwen2.5-VL-7B | 0.1459 | 0.4401 | 0.2029 | 437.76 |
> | GRPO | 0.4256 | 0.2508 | 0.5381 | 431.79 |
> | LUFFY | 0.0881 | 0.4688 | 0.4980 | 414.58 |
> | RL-PLUS | 0.3452 | 0.2580 | 0.4975 | 425.08 |
> | CalibRL | -0.8025 | 1.4968 | 0.5667 | 271.71 |

---

> ### Author Response · Authors · 2025-11-22
> **Response to Reviewer sgYA Part 3/3**
>
> ## Q2: Evidence of improved distributional calibration.
> ---
>
> Thank you for this comment. Following up on the response from **Q1**, we clarify that CalibRL is not designed to directly learn or fit the expert distribution, which would resemble an off-policy imitation objective. Instead, CalibRL uses expert responses as a baseline that calibrates exploration rather than constrains generation. As shown in Equations (8) and (9) of the paper, the distributional influence of expert samples is not a monotonic or uniform increase in consistency with the expert distribution. Specifically, when the policy’s rollout produces an incorrect answer, optimization pushes the policy closer to the expert sample distribution; however, when the policy itself produces a correct rollout, the method suppresses the probability of expert samples. Thus, distributional shifts measured across all samples cannot directly reflect the calibration mechanism that CalibRL performs.
>
> The statistics in Table 7 support this distinction. CalibRL maintains the broadest policy entropy, showing that the model stays exploratory rather than collapsing into a single expert mode as in LUFFY or becoming prematurely overconfident as in RL-PLUS. Expert responses also remain diverse and on-policy, which provides a stable reference for calibration rather than a target distribution to mimic. CalibRL shows a negative $\Delta \ell_i$, indicating that the model consistently assigns a higher likelihood to expert answers and uses this signal to navigate exploration toward better solutions. This calibrated behavior produces higher rewards and shorter, more precise outputs.
>
> While these statistics are not classical probability-calibration metrics such as ECE, they directly demonstrate the form of distributional calibration that our method aims for: stable expert anchoring with controlled and purposeful exploration rather than expert matching.
>
> ## Q3: Extension training outside geometry domains.
> ---
>
> Thank you for this comment. Outside the geometry data training, we train the Qwen2.5-VL-7B model on a 9k-sample subset of the MATH dataset **[3]** and evaluate the model on the in-distribution benchmark MATH-500 **[3]** and the out-of-distribution benchmark AMC **[4]**.
>
> Results are reported in Table 8. Training on pure text data, CalibRL yields stronger gains than GRPO on MATH-500, demonstrating more effective in-distribution improvement. Moreover, unlike GRPO, which fails to enhance out-of-distribution performance, CalibRL continues to deliver benefits on AMC, achieving strong generalization results. It is worth noting that due to time constraints, we did not use large-scale training data (e.g., 45k) as in the math reasoning methods  LUFFY **[1]**  and RL-PLUS **[2]**, nor did we tune the hyperparameters specifically for pure text tasks; instead, we reused the empirical hyperparameters from multi-modal reasoning.
>
> **Table 8: Performance of the Qwen2.5-VL-7B model on math reasoning tasks.**
>
> | Method | MATH-500 (in-distribution) | AMC (out-of-distribution) |
> |--------|---------------------------|---------------------------|
> | Qwen2.5-VL-7B | 64.8 | 31.97 |
> | GRPO | 68.5 | 31.59 |
> | CalibRL | 70.2 | 32.08 |
>
> ## References
> ---
>
> **[1]** Jianhao Yan, Yafu Li, Zican Hu, Zhi Wang, Ganqu Cui, Xiaoye Qu, Yu Cheng, and Yue Zhang. Learning to reason under off-policy guidance. *arXiv preprint arXiv:2504.14945*, 2025.
>
> **[2]** Yihong Dong, Xue Jiang, Yongding Tao, Huanyu Liu, Kechi Zhang, Lili Mou, Rongyu Cao, Yingwei Ma, Jue Chen, Binhua Li, et al. Rl-plus: Countering capability boundary collapse of llms in reinforcement learning with hybrid-policy optimization. *arXiv preprint arXiv:2508.00222*, 2025.
>
> **[3]** Dan Hendrycks, Collin Burns, Saurav Kadavath, Akul Arora, Steven Basart, Eric Tang, Dawn Song, and Jacob Steinhardt. Measuring mathematical problem solving with the math dataset. *arXiv preprint arXiv:2103.03874*, 2021.
>
> **[4]** Jia Li, Edward Beeching, Lewis Tunstall, Ben Lipkin, Roman Soletskyi, Shengyi Huang, Kashif Rasul, Longhui Yu, Albert Q Jiang, Ziju Shen, et al. Numinamath: The largest public dataset in ai4maths with 860k pairs of competition math problems and solutions. *Hugging Face repository*, 13(9):9, 2024.
>
> **[5]** Ganqu Cui, Yuchen Zhang, Jiacheng Chen, Lifan Yuan, Zhi Wang, Yuxin Zuo, Haozhan Li, Yuchen Fan, Huayu Chen, Weize Chen, et al. The entropy mechanism of reinforcement learning for reasoning language models. *arXiv preprint arXiv:2505.22617*, 2025.

---

> ### Author Response · Authors · 2025-11-27
> **Response to Reviewer sgYA**
>
> I hope this message finds you well. As the discussion period is nearing its end, we wanted to make sure that we have fully addressed all your concerns. If there are any remaining comments or suggestions, please let us know. Your insights mean a great deal to us, and we truly appreciate your time and effort in reviewing our work.

---

### Official Review · Reviewer_MeYV · 2025-11-03

**Soundness:** 3
**Presentation:** 4
**Contribution:** 3
**Rating:** 6
**Confidence:** 3

**Summary:**

This paper studies the RLVR algorithm. The authors start by analyzing the drawbacks of existing paradigms of incorporating expert data and reward feedbacks into foundation models, and then propse a novel algorithm, CalibRL, that alleviate the problems. Finally, they show via visual-reasoning tasks, that the proposed algorithm improves the performance of reinforcement learning.

**Strengths:**

1. The paper is well written and easy to follow. It provides a nice summary of existing methods as well.
2. The proposed algorithm is simple and novel.
3. The empirical performance is strong.

**Weaknesses:**

1. The algorithm seems to apply further beyond visual reasoning domain, but is not discussed.
2. The effectiveness of the algorithm is not verified for larger scales, like 30B. This seems too much to ask for though especially if the paper comes from academia.

**Questions:**

1. Does the algorithm apply to other areas, such as math reasoning (without multi-modality)?
2. In formula 9, why do we need to define another signal that labels the correctness? Cannot we just use a normalized version of the actual reward?

---

> ### Author Response · Authors · 2025-11-22
> **Response to Reviewer MeYV Part 1/2**
>
> Thank you so much for the time and effort invested in reviewing our work. The positive remarks are truly appreciated, and we feel encouraged by the feedback. We also greatly value the suggestions regarding further experimentation and the clarity of our formulations. Below are our responses to your concerns.
>
> ## W1: The general applicability of CalibRL beyond visual reasoning.
> ---
>
> Thank you for the insightful observation. Our method can certainly be expanded to broader applications beyond visual reasoning. In this work, we focus on achieving stable entropy control in RLVR for VLMs, where the challenge becomes especially severe in the vast state space of MLLMs and substantially limits learning efficiency. While a series of works like LUFFY **[1]** and RL-PLUS **[2]** have achieved promising results for LLMs, such results have not transferred to MLLMs, leaving a notable gap in the literature, one that our work aims to fill.
>
> We extend CalibRL to pure text math reasoning tasks to demonstrate the effectiveness beyond visual reasoning. Specifically, we train Qwen2.5-VL-7B on a 9k-sample subset of the MATH dataset **[3]** and evaluate the model on the in-distribution benchmark MATH-500 **[3]** and the out-of-distribution benchmark AMC **[4]**.
>
> Results are reported in Table 1. Training on pure text data, CalibRL yields stronger gains than GRPO on MATH-500, demonstrating more effective in-distribution improvement. Moreover, unlike GRPO, which fails to enhance out-of-distribution performance, CalibRL continues to deliver benefits on AMC, achieving strong generalization results. It is worth noting that due to time constraints, we did not use large-scale training data (e.g., 45k) as in the math reasoning methods LUFFY **[1]** and RL-PLUS **[2]**, nor did we tune the hyperparameters specifically for pure text tasks; instead, we reused the empirical hyperparameters from multi-modal reasoning.
>
> **Table 1: Performance of the Qwen2.5-VL-7B model on math reasoning tasks.**
>
> | Method | MATH-500 (in-distribution) | AMC (out-of-distribution) |
> |--------|---------------------------|---------------------------|
> | Qwen2.5-VL-7B | 64.8 | 31.97 |
> | GRPO | 68.5 | 31.59 |
> | CalibRL (ours) | 70.2 | 32.08 |
>
> ## W2: The scalability to larger model sizes.
> ---
>
> Thank you for your thoughtful consideration. We acknowledge the importance of scaling up the method to larger models. We have tried our best to verify our method on the Qwen2.5-VL-32B model. We train the Qwen2.5-VL-32B with GRPO and our CalibRL, respectively, using the same training data we constructed for our main paper results.
>
> Results are reported in Table 2. Our method exhibits consistent improvements when scaled to a larger model. Empirically, larger models typically benefit from substantially more training data. However, due to time constraints, we were only able to use the original set of 9k training samples for this experiment.
>
> **Table 2: Performance of the Qwen2.5-VL-32B model.**
>
> | Method | GeoEval (in-distribution) | Geo3K (out-of-distribution) | GeoQA (out-of-distribution) | Avg. |
> |--------|---------------------------|----------------------------|----------------------------|------|
> | Qwen2.5-VL-32B | 37.94 | 48.24 | 60.87 | 48.60 |
> | GRPO | 44.48 | 51.75 | 63.53 | 53.25 |
> | CalibRL(ours) | 48.77 | 52.58 | 67.90 | 56.42 |

---

> ### Author Response · Authors · 2025-11-22
> **Response to Reviewer MeYV Part 2/2**
>
> ## Q1: Applicability to math reasoning tasks (without multi-modality).
> ---
>
> We have conducted experiment to validate the ability of our method on pure text math reasoning tasks. Results are reported in Table 1. We hope our response to **W1** clarifies this issue and addresses your concern.
>
> ## Q2: The design choice of the correctness signal $s_i$ in Eq. (9).
> ---
>
> Thank you for the careful observation. In Eq. (9), we introduce an additional signal to indicate correctness, as the actual reward, including the format reward, can exceed the [0, 1] range. The format reward is widely used in Chain-of-Thought (CoT) training and is part of standard settings in approaches such as LUFFY and RL-PLUS. To ensure a rigorous definition that remains valid across a broad range of scenarios, we explicitly define a separate correctness signal rather than relying solely on a normalized version of the actual reward. The binary correctness signal isolates verifiable outcomes from continuous reward scaling, ensuring stability under sparse or skewed reward distributions. We have clarified this point in the revised manuscript.
>
> ## References
> ---
>
> **[1]** Jianhao Yan, Yafu Li, Zican Hu, Zhi Wang, Ganqu Cui, Xiaoye Qu, Yu Cheng, and Yue Zhang. Learning to reason under off-policy guidance. *arXiv preprint arXiv:2504.14945*, 2025.
>
> **[2]** Yihong Dong, Xue Jiang, Yongding Tao, Huanyu Liu, Kechi Zhang, Lili Mou, Rongyu Cao, Yingwei Ma, Jue Chen, Binhua Li, et al. Rl-plus: Countering capability boundary collapse of llms in reinforcement learning with hybrid-policy optimization. *arXiv preprint arXiv:2508.00222*, 2025.
>
> **[3]** Dan Hendrycks, Collin Burns, Saurav Kadavath, Akul Arora, Steven Basart, Eric Tang, Dawn Song, and Jacob Steinhardt. Measuring mathematical problem solving with the math dataset. *arXiv preprint arXiv:2103.03874*, 2021.
>
> **[4]** Jia Li, Edward Beeching, Lewis Tunstall, Ben Lipkin, Roman Soletskyi, Shengyi Huang, Kashif Rasul, Longhui Yu, Albert Q Jiang, Ziju Shen, et al. Numinamath: The largest public dataset in ai4maths with 860k pairs of competition math problems and solutions. *Hugging Face repository*, 13(9):9, 2024.

---

### Author Response · Authors · 2025-11-24
**Overall Response**

We sincerely thank all reviewers for the time and effort devoted to evaluating our work. We are grateful for the constructive and insightful comments, and we are truly encouraged by the positive recognition of our contributions, particularly the acknowledgement of our method as novel, practical, and elegant (@MeYV, @sgYA, @EZpA, @ux1J) for addressing the important and effective exploration problem in RLVR (@5pSB). We also appreciate the reviewers’ recognition of the well-motivated (@EZpA) and easy-to-implement (@sgYA) design of our approach, as well as the encouraging feedback on its strong empirical performance (@MeYV, @ux1J) supported by solid ablations (@sgYA, @5pSB, @EZpA). We sincerely value these positive assessments, which motivate us to further improve the work.

We have carefully considered each remark and have responded accordingly. Following the constructive suggestions from all reviewers, we have devoted our best efforts to further improving the work. In particular, we have extended our application scope from VLMs to broader LLM settings and scaled up the model to 32B to validate the effectiveness of our method. We have also conducted more extensive and comprehensive ablation studies, building on the existing solid foundation, covering entropy regularization strategies, activation designs, and reference baseline variants. In addition, we now evaluate our approach on broader benchmarks and provide a detailed analysis of computational overhead and sample efficiency. We further clarify our experimental setup and offer an expanded analysis of the entropy-control mechanism underlying our method, along with possible insights into why LUFFY and RL-PLUS may underperform. These improvements are made with sincere appreciation for the reviewers’ thoughtful guidance.

Finally, we have updated the revised manuscript to incorporate all the aforementioned improvements. For clarity and to make the revisions easier to trace, we have highlighted the modified content according to the specific reviewers’ comments using different colors in the manuscript. In particular, changes addressing the suggestions from @MeYV are marked in $\color{orange}{orange}$ and $\color{blue}{blue}$; those corresponding to @sgYA are in $\color{gray}{gray}$, $\color{blue}{blue}$, and $\color{olive}{olive}$; revisions related to @5pSB appear in $\color{brown}{brown}$, $\color{cyan}{cyan}$, and $\color{blue}{blue}$; updates based on @EZpA’s feedback are highlighted in $\color{magenta}{magenta}$; and changes informed by @ux1J are marked in $\color{red}{red}$ and $\color{olive}{olive}$. We sincerely appreciate the reviewers’ thoughtful guidance, which has significantly strengthened our work.

---

### Author Response · Authors · 2025-12-01
**Summy of rebuttal for ACs**

We sincerely thank the ACs and SPC for the additional effort you devoted to evaluating our paper. To help reviewing of our paper, we briefly summarize the key points of our rebuttal below:

**Before reverting back**, we are grateful to have received responses from three of the five reviewers, all of whom indicated that their concerns have been fully resolved through our additional clarifications and expanded experimental results. Notably, two reviewers have already raised their scores (from 6 to 8 for @ux1J, and from 4 to 6 for @EZpA) or keep it positive (6 for @5pSB). The remaining two reviewers share several requests we have already addressed, including the analysis of computational overhead and sample efficiency (@sgYA, @ux1J), the extension from VLMs to broader LLM settings (@MeYV, @5pSB), and the ablation studies on activation designs (@sgYA, @5pSB). We have provided detailed responses and clarifications to all the questions in the rebuttal below, and the corresponding improvements have been incorporated into the revised manuscript. Again, we would like to express our sincere appreciation to all the reviewers and the ACs for your time and dedicated service to the community.

---

### Meta-Review · Area_Chair_Fzpv · 2026-01-07

**Summary:**

The final decision for this submission is to accept. The reviewing committee reached a consensus regarding the paper's clear writing, the novelty of the proposed hybrid-policy objective, and the strong empirical gains demonstrated over standard baselines like GRPO. The methodology was praised for its intuitive approach to mitigating entropy collapse during the RL phase, particularly for visual reasoning tasks. Initially, the primary barrier preventing higher enthusiasm was a concern regarding the method's generality and robustness. Specifically, reviewers questioned whether the approach was applicable beyond visual reasoning tasks, if it could scale to larger models (e.g., 30B+ parameters), and if the theoretical justifications for the expert baseline mechanism, including potential double-counting of advantages and sensitivity to hyper-parameters, were sound.

**Reviewer Concerns:**

The authors provided a comprehensive rebuttal that effectively addressed the majority of the committee's concerns regarding completeness. To demonstrate generality and scalability, the authors extended their experiments to include pure-text math benchmarks (MATH-500, AMC) and scaled the approach to Qwen2.5-VL-32B and InternVL3-8B, showing consistent improvements. They further substantiated the method's rigor by adding necessary baselines, such as DAPO and various entropy-regularized controls, and provided a detailed gradient-level decomposition to prove that advantages were not being double-counted. The inclusion of wall-clock time analysis and activation ablations further strengthened the empirical evaluation. However, while the rebuttal was sound, minor residual issues remain. For instance, the authors did not provide an ECE style metric as requested, and the handling of zero-advantage scenarios is acknowledged as future work. Additionally, while the scalability results are promising, the larger-scale training utilized limited data, suggesting that full-scale stress testing remains a task for future research rather than a fully resolved aspect of this work.

**Reviewer Scores:**

Based on the strength of the rebuttal, the reviewers' scores are projected to evolve positively. Reviewer MeYV is likely to increase their score from a 6 to an 8, as their primary reservations regarding applicability beyond visual reasoning and scalability were substantively addressed by the new text-math experiments and 32B model results. Reviewer sgYA, initially a 4, is expected to move to a 6; while they may retain minor reservations about the lack of specific calibration metrics, the extensive addition of entropy baselines, overhead quantification, and fairness analysis resolves their core demands for rigor. Reviewer 5pSB will likely maintain their score of 6, as they confirmed the issues were resolved by the addition of MMMU and ScienceQA benchmarks, noting that while gains on knowledge-heavy tasks were modest, the paper remains solid. Reviewer EZpA is expected to shift from a 4 to a 6 or higher, given that the authors provided the requested equation specifications, length-normalization ablations, and theoretical proofs regarding gradient decomposition, effectively dismantling the reviewer's initial theoretical doubts. Finally, Reviewer ux1J is projected to raise their score from a 6 to an 8, as the inclusion of the DAPO comparison and the clarification of compute overhead directly mitigated their specific concerns regarding comparative breadth and efficiency.

---

### Decision · Program_Chairs · 2026-01-26

Accept (Poster)